# Consciousness-specific dynamic interactions of brain integration and functional diversity

Andrea I. Luppi[1,2], Michael M. Craig[1,2], Ioannis Pappas [1,2,8], Paola Finoia [1,3], Guy B. Williams[2,4], Judith Allanson[2,5], John D. Pickard[2,3,4], Adrian M. Owen[6], Lorina Naci[7], David K. Menon[1,4] & Emmanuel A. Stamatakis [1,2]*

Prominent theories of consciousness emphasise different aspects of neurobiology, such as the integration and diversity of information processing within the brain. Here, we combine graph theory and dynamic functional connectivity to compare resting-state functional MRI data from awake volunteers, propofol-anaesthetised volunteers, and patients with disorders of consciousness, in order to identify consciousness-specific patterns of brain function. We demonstrate that cortical networks are especially affected by loss of consciousness during temporal states of high integration, exhibiting reduced functional diversity and compromised informational capacity, whereas thalamo-cortical functional disconnections emerge during states of higher segregation. Spatially, posterior regions of the brain's default mode network exhibit reductions in both functional diversity and integration with the rest of the brain during unconsciousness. These results show that human consciousness relies on spatio-temporal interactions between brain integration and functional diversity, whose breakdown may represent a generalisable biomarker of loss of consciousness, with potential relevance for clinical practice.

[1] Division of Anaesthesia, School of Clinical Medicine, University of Cambridge, Addenbrooke's Hospital, Hills Rd, CB2 0SP Cambridge, UK. [2] Department of Clinical Neurosciences, School of Clinical Medicine, University of Cambridge, Addenbrooke's Hospital, Hills Rd, CB2 0SP Cambridge, UK. [3] Division of Neurosurgery, School of Clinical Medicine, University of Cambridge, Addenbrooke's Hospital, Hills Rd, CB2 0SP Cambridge, UK. [4] Wolfson Brain Imaging Centre, University of Cambridge, Cambridge Biomedical Campus (Box 65), CB2 0QQ Cambridge, UK. [5] Department of Neurosciences, Cambridge University Hospitals NHS Foundation, Addenbrooke's Hospital, Hills Rd, CB2 0SP Cambridge, UK. [6] The Brain and Mind Institute, Western Interdisciplinary Research Building, N6A 5B7 University of Western Ontario, London, ON, Canada. [7] Trinity College Institute of Neuroscience, School of Psychology, Lloyd Building, Trinity College Dublin, Dublin Dublin 2, Ireland. [8] Present address: Helen Wills Neuroscience Institute, 210 Barker Hall, University of California – Berkeley, 94720 Berkeley, CA, USA. *email: eas46@cam.ac.uk

The brain is a remarkably complex system, with activity patterns poised at a near-critical point between order and chaos[1], integrating inputs from different modalities into a unified experience of the world. Recent scientific theories of consciousness have appealed to these characteristics to explain how the diverse repertoire of human conscious experiences arises from brain function[2–7].

Specifically, consciousness is thought to require brain-wide information broadcasting by a "global workspace", whereby segregated component processes are integrated and made available for undertaking higher cognitive functions, producing a unitary experience[4,5]. Indeed, estimates of brain integration derived from graph theory[8] are reduced in brain networks under propofol anaesthesia[9–11] and in patients with disorders of consciousness[12].

In information theory, entropy quantifies the diversity or unpredictability of information content[13]. Using the entropy of different aspects of brain function to estimate the diversity of information, various studies have shown reduced entropy when consciousness fades, such as during sleep or anaesthesia[14–16], and more recently also in patients with disorders of consciousness[17]. Conversely, entropy increases during states of putatively enhanced consciousness induced by psychedelics (see Carhart-Harris (2018) for a recent review), supporting the idea that entropy, as measured from functional imaging, may reflect the richness and diversity of conscious experiences.

The influential integrated information theory identifies consciousness with a quantity $\Phi$, combining integration and diversity of information within a system[2,3]. Studies using a proxy for $\Phi$ based on the evoked EEG response to transcranial magnetic stimulation (TMS), known as perturbational complexity index, have been highly successful at discriminating between different states of consciousness, including anaesthesia and disorders of consciousness[18–21]. These studies demonstrate that diversity and integration are both relevant for consciousness, and raise the question of how they relate to each other neurobiologically. In particular, although the activity and connectivity of integrative hubs, such as the posterior cingulate/precuneus and other regions of the default mode network (DMN) are highly affected by loss of consciousness[22,23], a recent study reported that the entropy of the DMN and other higher-order cortices were only minimally affected by propofol anaesthesia[14].

To address this discrepancy, here we relate voxel-level measures of integration and entropy derived from resting-state functional MRI (rs-fMRI), to identify any regions showing consistent alterations in both quantities when consciousness is lost. If both entropy and integration are related to consciousness, then we should expect their alterations during unconsciousness to show some overlap in the brain, thereby identifying regions whose function is most relevant for consciousness.

Moreover, brain functions vary over time[24], exhibiting dynamics that have recently been shown to differ between states of consciousness in both anaesthetized non-human primates[11] and human patients with disorders of consciousness[17]. In awake healthy volunteers, the dynamics of brain functional connectivity have been shown to transition between states displaying higher segregation or integration[25,26], with the latter being related to higher arousal and improved cognitive performance. Thus, the interplay of brain entropy and integration in supporting consciousness may occur temporally as well as spatially, and we set out to investigate this hypothesis.

A further consideration is that there are multiple ways in which loss of consciousness may occur, whether through pharmacological interventions having widespread effects on brain function, or hypoxic-ischemic injuries affecting cortical and subcortical regions of the brain, or relatively localised traumatic brain injuries. In order to identify neurobiological signatures of loss of

consciousness that are generalisable across these multiple conditions, rather than being specific to any of them, here we investigate alterations in brain functions and dynamics during unconsciousness arising in common from all of the above-mentioned causes. Specifically, we compare brain measures of 16 healthy volunteers while awake and while undergoing deep anaesthesia with the common intravenous agent propofol, and we also compare the same 16 awake volunteers with 22 patients diagnosed with a disorder of consciousness (DOC; unresponsive wakefulness syndrome/vegetative state (UWS) or minimally conscious state, MCS) as a result of traumatic or hypoxic/ischemic brain injury.

Focusing on results that are common across the two datasets, we demonstrate that in both DOC patients and healthy volunteers undergoing propofol anaesthesia, measures of brain functional diversity and integrative capacity are related in both space and time, and their interactions are altered during unconsciousness. Furthermore, brain dynamics are also altered during unconsciousness: temporal states of high integration exhibit reduced integration and functional diversity in unconscious individuals, unlike predominantly segregated states. Additionally, thalamo-cortical disconnections become evident during temporal states of high segregation, whereas highly integrated states are primarily characterised by altered cortico-cortical connectivity; thus, our results contribute to reconcile discrepant findings in the literature about the relative roles of cortico-cortical and thalamo-cortical connectivity for consciousness[9,10,22,27–30]. Finally, we show that in both datasets, whole-brain connectivity and temporal entropy are both reduced in key regions of the brain's default mode network. By combining different biological approaches with diverse analytical methods investigating both spatial and temporal aspects of brain function, these results advance our understanding of conscious and unconscious states in the human brain, and their underlying brain dynamics.

## Results

**Reduced global integration and functional diversity**. To determine how integration and entropy interact spatially, we quantified the regional distribution of unconsciousness-related changes in these metrics across the brain. We first investigated how loss of consciousness affected the global functional connectivity of each region with the rest of the brain, which we took as reflecting its capacity for global integration. We estimated this with the Intrinsic Connectivity Contrast (ICC), a voxel-wise measure that uses graph theory to quantify each voxel's whole-brain connectivity, computed as the sum of the squared Pearson correlation between that voxel's timeseries and the timeseries of every other voxel in the brain[31].

Both localised decreases and increases in ICC were evident under deep propofol anaesthesia (Supplementary Fig. 1a and Supplementary Table 1), with prominent reductions in posterior cingulate/precuneus (PCC/PCU), medial prefrontal cortex (mPFC), left angular gyrus (AG) and left supramarginal gyrus (SMG) and medial temporal lobe (MTL). Increases were primarily localised in the bilateral caudate, and bilateral pre- and postcentral gyri.

Analogous results were observed when comparing the healthy volunteer cohort during their awake scan and the DOC patients (Supplementary Fig. 1b). ICC reductions were evident in PCC/PCU, left AG, mPFC, bilateral MTL, supramarginal gyri (SMG), opercular, postcentral and inferior parietal cortices, right fusiform and inferior temporal cortex (Supplementary Table 2). Increases were localised in the hippocampal formations, cerebellum, and bilateral caudate; the latter was shared with the propofol dataset (Supplementary Fig. 2 and Supplementary Table 3).

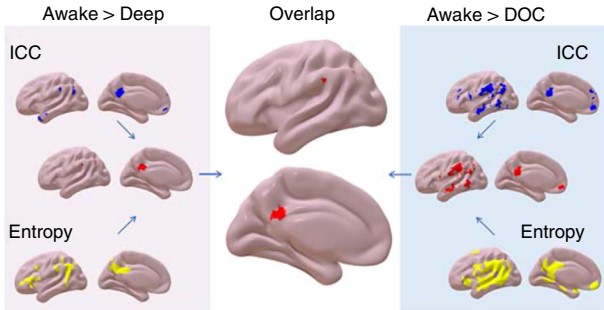

**Fig. 1** Brain maps of consciousness-related reductions in intrinsic connectivity contrast (ICC, reflecting integrative capacity) and sample entropy (reflecting functional diversity over time), and their overlaps between and within datasets. Regions in blue display reduced ICC during unconsciousness (increases are shown in Supplementary Figs. 1 and 2); yellow indicates reduced sample entropy during unconsciousness; and red indicates regions showing both reduced entropy and ICC. Overlaps between anaesthesia (left) and DOC (right) are shown in the middle. Images are shown on medial and lateral surfaces of a smoothed standard Montreal Neurological Institute (MNI-152) structural T1 scan, in neurological convention

We then quantified functional diversity as the sample entropy (SampEn) of voxelwise blood-oxygen-level-dependent (BOLD) signal timeseries[32] (see Methods). Sample entropy, which is derived from Approximate Entropy (ApEn)[33], is an approximation of Kolmogorov complexity for timeseries and is stable even for data sequences of limited length, such as fMRI timeseries[32]. SampEn quantifies how unpredictable a signal is, such that low values of SampEn would indicate that the signal is highly stereotyped - with a perfectly predictable series, such as [1,1,1,…] having a SampEn of zero, and SampEn increasing as the series becomes more disordered[32].

Widespread reductions in BOLD signal entropy were observed under anaesthesia, localised in the PCC/PCU, left AG extending to MTL and inferior parietal, and left middle and inferior frontal gyri. No increases were observed (Supplementary Fig. 1c and Supplementary Table 4). Crucially, reductions in BOLD signal entropy overlapped with ICC reduction due to anaesthesia in PCC/PCU, left AG and left SMG (Fig. 1, left panel). Likewise, DOC patients showed extensive sample entropy reductions in PCC/PCU and mPFC, as well as bilateral AG extending to inferior parietal, left inferior and middle and right middle and superior frontal cortices, and inferior and medial temporal lobes, including fusiform gyri and hippocampal formations (Supplementary Fig. 1d and Supplementary Table 5). Again, these entropy reductions largely overlapped with ICC reductions in our group of patients (Fig. 1, right panel).

Although sample entropy requires the choice of two parameters $m$ and $r$ (see Methods), entropy results were robust to parameter choices (Supplementary Fig. 3a, b). However, the use of a smaller smoothing kernel did result in fewer and smaller clusters of significantly different SampEn (Supplementary Fig. 3c, d). In particular, the left SMG cluster disappeared when comparing awake and anaesthetised individuals, and the left angular gyrus cluster was substantially reduced in size. Nevertheless, the precuneus cluster was maintained, despite the reduced size, further highlighting the specific importance of the default mode network.

In order to exclude the effect of subject motion as a possible confounding factor, we also compared the sample entropy of the head motion signals (three translations and three rotations) between the awake healthy controls and the two conditions of unconsciousness, using the same parameters as for the brain

entropy analysis described above. Although a significant reduction in the entropy of the head motion in the horizontal plane was observed when comparing DOC patients to controls (Supplementary Table 6), this was not the case for deep propofol anaesthesia, where instead an increase in the entropy of the timeseries of rotations around the vertical axis was observed, compared with the awake condition (Supplementary Table 7).

To characterise the consciousness-specific alterations of entropy and integration in the spatial domain, we then computed the overlaps between propofol- and brain injury-induced unconsciousness. The posterior cingulate/precuneus, left angular gyrus and left supramarginal gyrus showed reductions in both sample entropy and ICC, for both anaesthesia and disorders of consciousness (Fig. 1, central panel). To establish the spatial extent of connectivity changes we carried out follow-up seed-based analysis from these regions[31], which showed that during unconsciousness the connections between PCC and angular gyrus with the frontal portions of the DMN were attenuated, as were their anticorrelations with fronto-parietal control networks (FPN), in line with previous evidence[27,30,34–39]; reduced DMN-FPN anticorrelation was also observed from the SMG, which is part of the FPN (Supplementary Figs. 4–9).

**Dynamic functional connectivity alterations**. Since consciousness is believed to require a global workspace, integrating relevant information from separate modules[4,5], we then applied dynamic functional connectivity analysis[24] to identify temporal states of high integration and high segregation[25,26], and to investigate how loss of consciousness influenced their probability of occurrence and their functional diversity.

Following the approach of Shine et al.[25], the integrated and segregated states were identified by using a machine learning algorithm, known as k-means clustering, to separate subject- and session-specific sliding windows of dynamic functional connectivity (FC) into two clusters; these were then labelled as the "predominantly integrated" and "predominantly segregated" clusters, based on their graph-theoretical properties of participation coefficient and within-module degree Z-score[25], and each cluster was summarised by one centroid matrix of functional connectivity (see Methods).

Having computed the dynamic states of integration and segregation, we were able to investigate whether each state was similarly altered during unconsciousness induced by anaesthesia and injury, in terms of its pattern of functional connections (Methods). For comparison purposes, we also carried out these analyses for what we term "static" FC (spanning the entire scan duration; Fig. 2 and Supplementary Fig. 10).

In line with previous work, the overall picture from static FC revealed disconnections between the bilateral thalami and the precuneus[29], disconnections within DMN regions, reduced anticorrelations between DMN and FPN (showing as unconscious > conscious)[22,23], and within-FPN connectivity increases. Dynamic analysis of states that were partitioned as primarily integrated or primarily segregated revealed that alterations in cortico-cortical connectivity were primarily observed during the integrated state, whereas thalamo-cortical disconnections (left-lateralised) were specific to the predominantly segregated state (Fig. 2 and Supplementary Fig. 10).

**Temporal integration**. To identify the role of temporal integration in consciousness, we investigated how loss of consciousness affected the time spent in each state. For each individual, we computed the proportion of dynamic matrices assigned to each state, out of the total number (as there were only two states, the two analyses are complementary, and only one is reported).

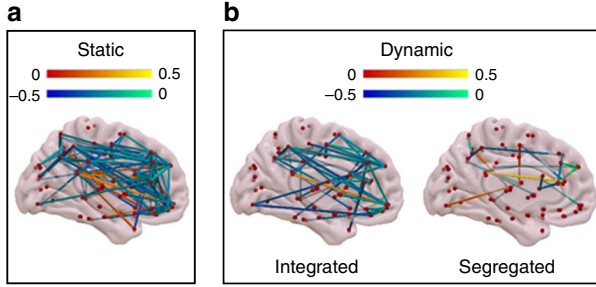

**Fig. 2** Common differences in functional connectivity (*r* values) when considering both awake volunteers > anaesthetized (repeated-measures *t*-test, FDR-corrected), and awake volunteers > DOC patients (two-samples *t*-test, FDR-corrected). **a** Differences from static FC (computed over the entire scanning length). **b** Differences from dynamic FC, observed only in the integrated state (left) or only in the segregated state (right). Hot colours indicate conscious > unconscious, and cold indicate unconscious > conscious (increased correlation, or decreased anticorrelation). Source data are provided as a Source Data file

Contrary to our expectation, time spent in the predominantly integrated state was not significantly different between awake ($M = 0.61$, $SD = 0.09$) and anaesthetized volunteers ($M = 0.55$, $SD = 0.12$), $t (15) = 1.79$, $g = 0.56$, $p = 0.093$ (repeated-measures *t*-test). Likewise, the awake volunteers did not significantly differ from the DOC patients ($M = 0.63$, $SD = 0.13$, $t (36) = -0.50$, $g = -0.16$, $p = 0.623$ (two-samples *t*-test; Supplementary Fig. 11).

**Optimal information capacity**. However, we reasoned that unconsciousness may also alter the network properties of the brain during each individual state of time-resolved functional connectivity. We used the graph-theoretical property of small-worldness to investigate this possibility (Methods). Small-world networks allow for cost-efficient organisation, by maximising local communication and minimising costly long-distance connections, thus balancing integration and segregation[40,41]. This measure is therefore often employed to estimate the optimality of a network's information capacity[11].

When considering FC calculated over the entire run, we observed a reduction in network small-worldness between the awake volunteers ($M = 2.21$, $SD = 0.22$) and DOC patients ($M = 1.97$, $SD = 0.36$, $t(36) = 2.31$, $g = 0.74$, $p = 0.028$, two-samples *t*-test); however, the healthy volunteers when anaesthetized did not show significant differences in small-worldness from when they were awake ($M = 2.04$, $SD = 0.27$, $t (15) = 2.05$, $g = 0.66$, $p = 0.063$, repeated-measures *t*-test). (Fig. 3, top left).

Nevertheless, adopting a dynamic approach revealed that the integrated state was characterised by reduced small-worldness for both anaesthetised healthy individuals ($M = 1.90$, $SD = 0.25$, $t (15) = 3.01$, $g = 1.03$, $p = 0.010$, repeated-measures *t*-test); and DOC patients ($M = 1.89$, $SD = 0.26$, $t (36) = 3.41$, $g = 1.10$, $p = 0.002$, two-samples *t*-test) when compared to consciousness ($M = 2.15$, $SD = 0.21$) (Fig. 3 middle left). Conversely, during the segregated state there was no significant difference in small-worldness between the awake individuals ($M = 2.07$, $SD = 0.24$) and either the same individuals under propofol anaesthesia, ($M = 1.97$, $SD = 0.24$, $t (15) = 1.11$, $g = 0.37$, $p = 0.286$, repeated-measures *t*-test) or the DOC patients ($M = 1.91$, $SD = 0.40$, $t (36) = 1.40$, $g = 0.45$, $p = 0.171$, two-samples *t*-test) (Fig. 3 bottom left). These results were robust to the choice of network node and edge definition, in both datasets—although reduced small-worldness was also observed during the segregated states, when employing liberal thresholds for edge definition (Supplementary Fig. 12 and Supplementary Tables 8–10).

**Entropy of temporal states**. We then investigated whether entropy also showed state-specific alterations during unconsciousness. Whereas voxelwise sample entropy (as discussed earlier) quantifies the unpredictability of the signal from each voxel over time, we also wished to examine a metric that better represented how these changes related to information exchange. To do this, we then tested levels of consciousness based on an alternative measure of brain entropy developed by Saenger et al.[42], which quantifies how diverse the pattern of connections of each brain region is, in terms of the mean normalised Shannon entropy of each region's distribution of FC values ("connectivity entropy"; see Methods).

For FC calculated over the entire run, significantly reduced mean connectivity entropy was observed between the awake ($M = 0.90$, $SD = 0.01$) and deep conditions ($M = 0.89$, $SD = 0.01$, $t (15) = 2.69$, $p = 0.017$, $g = 0.85$, repeated-measures *t*-test; Fig. 3, top right). Compared with the awake healthy controls, mean connectivity entropy was also reduced for the DOC patients ($M = 0.88$, $SD = 0.01$, $t (36) = 3.26$, $p = 0.002$, $g = 1.05$, two-samples *t*-test; Fig. 3, top right).

Network-specific investigations, based on the seven-network parcellation of Yeo et al.[43], were also performed to identify whether the observed entropy reductions were uniform across the brain, or localised to specific resting-state networks. This analysis revealed that the significant reduction in connectivity entropy under anaesthesia at whole-brain level was reflected by a reduction in DMN regions (Supplementary Fig. 13a and Supplementary Table 11). In DOC patients, the global effect corresponded to reductions in connectivity entropy of the DMN, but also FPN, dorsal attention, visual and limbic networks (Supplementary Fig. 13b and Supplementary Table 12).

Further state-specific analyses provided insight into the temporal origin of differences in connectivity entropy. Significantly reduced mean connectivity entropy was observed in the predominantly integrated state between participants when they were awake ($M = 0.904$, $SD = 0.007$) and anaesthetised ($M = 0.897$, $SD = 0.010$, $t(15) = 2.22$, $p = 0.046$, $g = 0.80$, repeated-measures *t*-test; Fig. 3, middle right). This global reduction was reflected by reduced connectivity entropy in the DMN (Supplementary Fig. 13c and Supplementary Table 13), in line with results from static FC.

Mean connectivity entropy in the predominantly integrated state was also reduced for DOC patients ($M = 0.89$, $SD = 0.01$, $t(36) = 3.43$, $p = 0.002$, $g = 1.10$, two-samples *t*-test) when compared with the awake volunteers (Fig. 3, middle right). The global reduction in connectivity entropy in the patient group reflected significant reductions in DMN, FPN and limbic networks (Supplementary Fig. 13d and Supplementary Table 14), reflecting results from static FC.

Conversely, no significant reductions in mean connectivity entropy of the predominantly segregated state were observed between awake ($M = 0.93$, $SD = 0.01$) and anaesthetised participants ($M = 0.92$, $SD = 0.01$, $t(15) = 1.43$, $g = 0.60$, $p = 0.174$, repeated-measures *t*-test), either globally or for specific subnetworks (Fig. 3, bottom right and Supplementary Fig. 13e and Supplementary Table 15). Thus, a marked contrast existed between the impact of anaesthesia on connectivity entropy in integrated and segregated states.

Instead, the mean connectivity entropy in the predominantly segregated state was significantly reduced when comparing the awake healthy controls with DOC patients ($M = 0.91$, $SD = 0.01$) $t(36) = 2.52$, $p = 0.014$, $g = 0.81$, two-samples *t*-test. (Fig. 3, bottom right). This reduction of global entropy in the segregated state of DOC patients was reflected in reductions in DMN, visual, somatomotor, dorsal attention and limbic networks (Supplementary Fig. 13f and Supplementary Table 16).

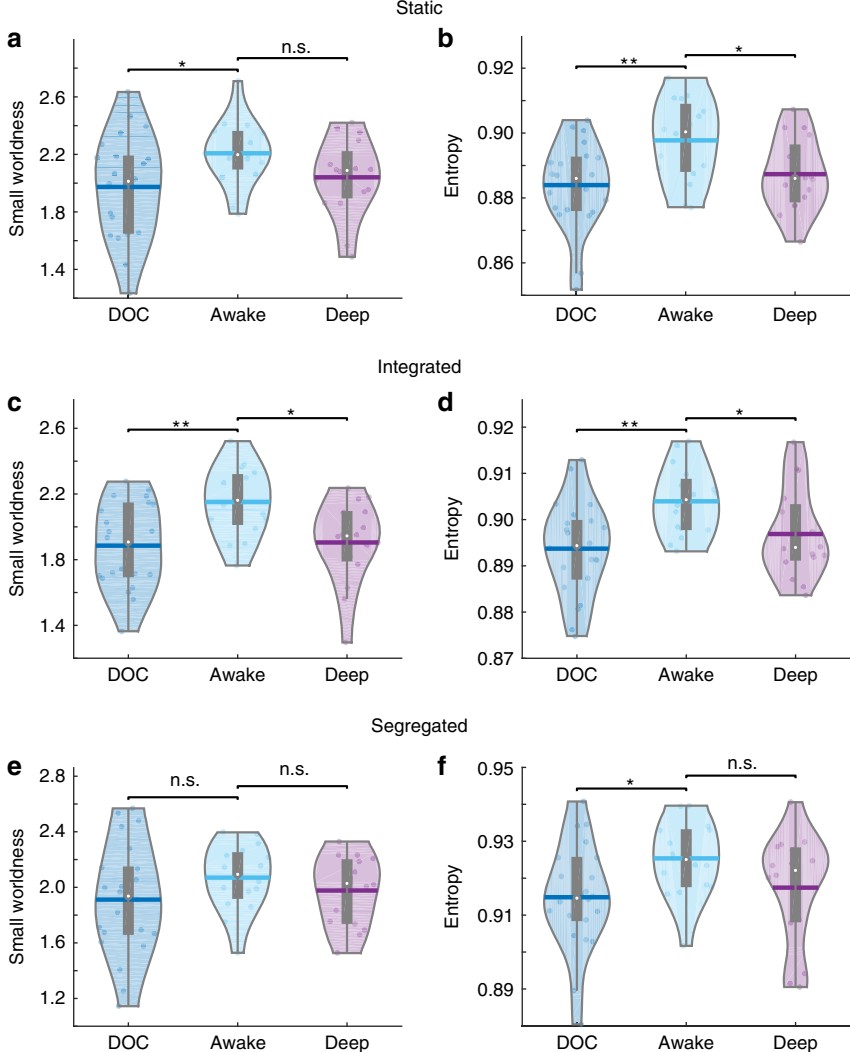

**Fig. 3** Violin plots of the mean connectivity entropy (left) and small-world values (right) for the static (**a**, **b**), integrated (**c**, **d**) and segregated (**e**, **f**) states, comparing conscious healthy controls and unconscious individuals due to anaesthesia (repeated_measures $t$-tests) and brain injury (two-samples $t$-tests). The small-world index was calculated as the ratio of normalised clustering coefficient to normalised characteristic path length. n.s. not significant; $*p < 0.05$; $**p < 0.01$; white circle, mean; centre line, median; box limits, upper and lower quartiles; whiskers, 1.5× interquartile range. Source data are provided as a Source Data file

**Comparison of different DOC aetiologies**. To assess whether our results for the DOC patients may be primarily driven by one particular aetiology (traumatic or hypoxic/ischemic brain injury), we repeated the analyses presented above, but including only the DOC patients, divided into two groups comparing DOC patients with traumatic brain injury (TBI; $N = 10$) and patients with hypoxic/ischemic injury (HBI; $N = 12$). No significant differences were observed between the two groups of DOC patients (Supplementary Table 17 and Supplementary Fig. 14).

## Discussion

We combined graph theory and dynamic functional connectivity to compare human fMRI data during conscious wakefulness, and during unconsciousness caused by anoxic brain injury, traumatic brain injury, and propofol anaesthesia. To identify general markers of loss of consciousness irrespective of its specific origin, we focused on brain alterations that were common to both datasets. Our results reveal that consciousness relies on the interaction between functional diversity as measured by entropy and integration, in both the spatial and temporal domains.

Spatially, we show that key regions of the default mode network, especially the posterior cingulate/precuneus, exhibit concurrent reductions in integrative capacity and functional diversity, estimated by the intrinsic connectivity contrast (in line with the ICC reductions under sevoflurane reported in other studies[31]) and sample entropy of BOLD signal timeseries, respectively. The ICC measures the integrative capacity of a given brain region because it reflects changes in either the inputs the region receives, or the outputs it broadcasts—both of which are crucial aspects of the process of integrating information. Being based on functional connectivity, the ICC is agnostic with regard to directionality; however, computational modelling has shown that nodes with high degree, to which ICC is related, tend to be the target of information flow from nodes with lower degree[44]. Thus, our results may suggest that during unconsciousness, cortical hubs in the posterior DMN regions receive less information from the rest of the brain, and consequently exhibit reduced entropy, reflecting reduced informational content and diversity. Since loss of consciousness is expected to compromise the ability to exchange and integrate information from across the cortex in the brain's global workspace, our results support the identification

of the latter at least in part with the DMN. They also support the predictions of integrated information theory, according to which unconsciousness can result from a loss of information and diversity as well as integrative capacity[2,3].

In the temporal domain, the present work advances our knowledge by demonstrating the dynamic nature of DMN involvement in loss of consciousness. For both DOC and anaesthesia the predominantly integrated state, which is related to higher cognitive performance and alertness[25], was characterised by reduced DMN-FPN anticorrelations. Conversely, disconnections between DMN and thalamus were specific to the predominantly segregated state, and within-DMN connectivity reductions were observed in both states. Thus, the results presented herein refine our understanding of the relative importance of thalamo-cortical and cortico-cortical connectivity for consciousness,[9,10,22,27–30] by revealing that alterations of DMN cortico-cortical and thalamo-cortical connections appear to vary dynamically, depending on the brain's state of integration or segregation.

Dynamic FC analysis also revealed interactions between functional diversity and integration in the temporal dimension: the predominantly integrated state appears particularly vulnerable to consciousness-related reductions in the mean Shannon entropy of each region's pattern of functional connectivity. Although this state was not visited less often during unconsciousness, we did find reduced small-worldness, a measure of information capacity, suggesting that these qualities of the integrated state may be more relevant for consciousness than time spent in it, at least in the context of resting-state data. It is important to bear in mind that applications of small-worldness and other graph-theoretical measures to brain networks are inherently limited by the noisiness of imaging modalities, such as fMRI[45]—a concern that we sought to address in this work by replicating our results pertaining to small-worldness across multiple thresholds, and with different node and edge definitions for our networks.

The reductions in brain entropy that we observed are consistent with the recent Entropic Brain Hypothesis[6,7] and with reports of reduced entropy with diminished consciousness[15,16,46], including using dynamic functional connectivity[17], but they appear in partial contrast with those of Liu et al.[14]. These authors reported that entropy diminishes in primary sensory systems when transitioning from light to deep propofol sedation, but not in DMN or other higher cognitive systems. However, the present reductions in DMN entropy were obtained from two different measures (spatial and temporal), in both anaesthesia and DOC patients, demonstrating their robustness. Moreover, it has been shown that at propofol doses lower than ~2.7 µg mL$^{-1}$, which is when EEG slow-wave activity reaches saturation (SWAS)[28,47], the brain is still responsive to external stimuli, despite behavioural unresponsiveness, and thus perceptual awareness is thought to be preserved[28]. Although SWAS cannot be confirmed, since EEG monitoring was not present, the dose of propofol we used was substantially higher than the one used by Liu et al.[14], and closer to SWAS-inducing level. Thus, preserved and reduced DMN entropy in the two studies may reflect preserved and compromised perceptual awareness, respectively—thereby resolving the apparent discrepancy, in line with extensive evidence relating DMN alterations to unconsciousness, for both anaesthesia[22,27,30,38,48–50] and DOC[23,34–37,39,51–54].

In the present study, the differences in network properties under anaesthesia—concurring with those observed in patients with DOC—only became apparent in the integrated state of dynamic FC, whereas static FC (i.e., computed over the entire scanning duration) failed to detect them. Indeed, earlier studies using only static FC analysis even reported increased small-worldness under propofol[9,55] whereas reduced small-worldness

under propofol was later demonstrated using dynamic FC in macaques, consistent with our results[11]. In line with recent work combining graph theory and dynamic functional connectivity to study DOC patients[17], our results further demonstrate the importance of time-resolved analyses in the study of consciousness, to uncover otherwise hidden similarities between different datasets.

Importantly, this work dealt exclusively with data from resting state scanning, which is known to preferentially recruit the DMN[56]; there is evidence that under different paradigms other networks, such as the auditory and FPN, are able to distinguish between conscious and unconscious states[57]. Further work could also relate the present results to independent markers of consciousness, such as SWAS[28] or naturalistic paradigms[57], and determine whether they can discriminate between different levels of sedation, or different disorders of consciousness.

Indeed, a limitation of this work is that we have implicitly assumed that both individuals under deep propofol anaesthesia and UWS and MCS patients are unconscious. However, MCS patients owe their classification to occasional signs of volitional behaviour, which may reflect minimal levels of consciousness[58]. Furthermore, disorders of consciousness are prone to relatively high rate of misdiagnosis, with patients categorized as UWS subsequently exhibiting signs of awareness when more sensitive measures are employed[57]. Adding to this complication, dreaming has been reported during anaesthesia in up to 27% of cases[59]. Consequently, despite lack of behavioural responsiveness it is not possible to say conclusively that all individuals examined here were completely unconscious, in the sense of having no subjective experiences. Using additional markers of consciousness, such as SWAS[28], PCI[18] or naturalistic paradigms[57] may be required in future studies to provide additional evidence of unconsciousness independent of behavioural responsiveness.

Other issues of consideration include the scanner hardware and acquisition parameters, which were not identical for the two cohorts discussed in the article. A further confound may be that DOC patients had reduced entropy of motion timeseries in the horizontal plane compared to awake controls, whereas anaesthesia led to increased entropy in the rotation around the vertical axis. Additionally, it is important to consider how the anaesthetic agent can indirectly affect measures of brain activity by altering physiological parameters, such as arterial concentration of carbon dioxide. Since anaesthetized subjects in this study were not intubated but rather spontaneously ventilating, propofol-induced respiratory depression may have resulted in hypoventilation and increased arterial $CO_2$ levels. We did not measure end-tidal or arterial $CO_2$ concentrations in our subjects. However, propofol only has relatively minor effects on brain hemodynamics[60], and BOLD fluctuation amplitudes, connectivity strength and the spatial extent of connectivity maps have been shown to be unaffected by hypercapnia during the resting state in rats[61], thus mitigating this concern.

In addition to noting that both motion and cardiac, respiratory and physiological noise artefacts are accounted for in our denoising procedure (see Methods), we believe that these concerns should be further mitigated by our decision to adopt a comparative approach in this work, focusing only on results that were observed in both DOC and anaesthesia: if any of the results we observed when comparing DOC patients and controls had been due solely to the differences in acquisition and scanning parameters, or subject motion, such results should not have also been observed under conditions of anaesthesia, where those confounds were not present thanks to the within-subjects design—and vice versa for the concern about hypercapnia during anaesthesia. Moreover, using the same group of awake healthy volunteers as controls for both states of unconsciousness helps

to ensure that the results we have found are not confounded by differences in control groups, but rather represent deviations from the same common baseline. Therefore, we expect that the common results we report should represent genuine markers of loss of consciousness, rather than reflecting any idiosyncratic aspects of the specific datasets used here.

Overall, we have demonstrated that loss of consciousness—whether due to propofol anaesthesia, hypoxic/ischemic brain injury, or traumatic brain injury—is accompanied by reduced functional diversity and integrative capacity in the posterior DMN, especially during temporal states of high integration. Additionally, our time-resolved analysis led us to the discovery that the relative importance of cortico-cortical and thalamo-cortical connectivity in supporting consciousness varies dynamically with the brain's state of integration or segregation, thereby clarifying a previous point of controversy in the literature. The effects reported were observed in both DOC patients (with no differences between TBI and HBI patients) and healthy volunteers undergoing propofol anaesthesia. This replication increases the robustness of our results, and it allowed us to narrow down on properties of brain function that are likely to be consciousness-specific, whose alterations may, therefore, represent general neurobiological signatures of loss of consciousness, with potential translational value for the detection of consciousness during surgery or in patients with a diagnosis of vegetative state/unresponsive wakefulness syndrome.

## Methods

**Volunteer recruitment for propofol data collection**. The propofol data were collected at the Robarts Research Institute in London, Ontario (Canada) between May and November 2014. A total of 19 (18–40 years; 13 males) healthy, right-handed, native English speakers, with no history of neurological disorders were recruited. The Health Sciences Research Ethics Board and Psychology Research Ethics Board of Western University (Ontario, Canada) ethically approved this study, and all relevant ethical guidelines were followed. Each volunteer provided written informed consent, and received monetary compensation for their time. Due to equipment malfunction or physiological impediments to anaesthesia in the scanner, data from three participants (1 male) were excluded from analyses, leaving 16.

**Procedure and design for propofol data collection**. Resting-state fMRI data were acquired at no sedation (Awake), and Deep sedation (anaesthetised: Ramsay score of 5[62]). Ramsay level was independently assessed by two anaesthesiologists and one anaesthesia nurse in the scanning room before fMRI acquisition began, in each condition. Additionally, participants performed two tests: a computerised auditory target-detection task and a memory test of verbal recall, to evaluate their level of wakefulness independently of the assessors.

For the Awake condition, participants did not receive a Ramsey score, as this scale is designed for patients in critical care. Instead, they had to be fully awake, alert and communicating appropriately. An infrared camera located inside the scanner was used to monitor wakefulness. For the Deep sedation condition, propofol was administered intravenously using an AS50 auto syringe infusion pump (Baxter Healthcare, Singapore); step-wise sedation increments sedation were achieved using an effect-site/plasma steering algorithm combined with the computer-controlled infusion pump. Further manual adjustments were performed as required to reach target concentrations of propofol, as predicted by the TIVA Trainer (European Society for Intravenous Anaesthesia, eurosiva.eu) pharmacokinetic simulation program. This software also specified the blood concentrations of propofol, following the Marsh 3-compartment model, which were used as targets for the pharmacokinetic model providing target-controlled infusion. The initial propofol target effect-site concentration was $0.6 \ \mu g \ mL^{-1}$, with oxygen titrated to maintain SpO2 above 96%. Concentration was then increased by increments of $0.3 \ \mu g \ mL^{-1}$, and Ramsay score was assessed: if lower than 5, a further increment occurred. Participants were deemed to have reached Ramsay level 5 once they stopped responding to verbal commands, were unable to engage in conversation, and were rousable only to physical stimulation. Data acquisition began once loss of behavioural responsiveness occurred for both tasks, and the three assessors agreed that Ramsay sedation level 5 had been reached.

The mean estimated effect-site and plasma propofol concentrations were kept stable by the pharmacokinetic model delivered via the TIVA Trainer infusion pump; the mean estimated effect-site propofol concentration was 2.48 (1.82–3.14) $\mu g \ mL^{-1}$, and the mean estimated plasma propofol concentration was 2.68 (1.92–3.44) $\mu g \ mL^{-1}$. Mean total mass of propofol administered was 486.58 (373.30–599.86) mg. These values of variability are typical for the pharmacokinetics

and pharmacodynamics of propofol. Since the sedation procedure did not take place in a hospital setting, airway security could not be ensured by intubation during scanning, although two anaesthesiologists closely monitored each participant. Consequently, scanner time was minimised to ensure return to normal breathing following deep sedation. No state changes or movement were noted during the deep sedation scanning for any of the participants included in the study.

In the scanner, subjects were instructed to relax with closed eyes, without falling asleep; 8 min of fMRI scan without any task ("resting-state") were acquired for each participant. Additionally, a separate 5-min long scan was also acquired while a plot-driven story was presented through headphones to participants, who were instructed to listen while keeping their eyes closed. The present analysis focuses on the resting-state data only; the story scan data are not relevant to the work presented here, and will not be discussed further.

**Propofol MRI data acquisition**. MRI scanning was performed using a 3-Tesla Siemens Tim Trio scanner (32-channel coil), and 256 functional volumes (echo-planar images, EPI) were collected from each participant, with the following parameters: slices = 33, with 25% inter-slice gap; resolution = 3 mm isotropic; TR = 2000 ms; TE = 30 ms; flip angle = 75 degrees; matrix size = 64 × 64. The order of acquisition was interleaved, bottom-up. Anatomical scanning was also performed, acquiring a high-resolution T1-weighted volume (32-channel coil, 1 mm isotropic voxel size) with a 3D MPRAGE sequence, using the following parameters: TA = 5 min, TE = 4.25 ms, 240 × 256 matrix size, 9 degrees FA.

**Preprocessing of propofol fMRI data**. We preprocessed the functional imaging data using a standard pipeline, implemented within the SPM12-based (http://www.fil.ion.ucl.ac.uk/spm) toolbox CONN (http://www.nitrc.org/projects/conn), version 17f[63]. The pipeline comprised the following steps: removal of the first five scans, to allow magnetisation to reach steady state; functional realignment and motion correction; slice-timing correction to account for differences in time of acquisition between slices; identification of outlier scans for subsequent regression by means of the quality assurance/artefact rejection software ART (http://www.nitrc.org/projects/artifact_detect); spatial normalisation to Montreal Neurological Institute (MNI-152) standard space with 2 mm isotropic resampling resolution, using the segmented grey matter image from each volunteer's high-resolution T1-weighted image, together with an a priori grey matter template; spatial smoothing with a Gaussian kernel of 6 mm full width at half-maximum (FWHM).

**Denoising of propofol fMRI data**. To reduce noise due to cardiac and motion artefacts, we applied the anatomical CompCor method of denoising the functional data[64], also implemented within the CONN toolbox. The anatomical CompCor method involves regressing out of the functional data the following confounding effects: the first five principal components attributable to each individual's white matter signal, and the first five components attributable to individual cerebrospinal fluid (CSF) signal; six subject-specific realignment parameters (three translations and three rotations) as well as their first- order temporal derivatives; the artefacts identified by ART; and main effect of scanning condition[64]. Linear detrending was also applied, and the subject-specific denoised BOLD signal timeseries were band-pass filtered to eliminate both low-frequency drift effects and high-frequency noise, thus retaining frequencies between 0.008 and 0.09 Hz.

**Recruitment of DOC patients**. A sample of 71 patients with disorders of consciousness was included in this study. Patients were recruited from specialised long-term care centres. To be invited to the study, patients must have had a DOC diagnosis, written informed consent to participation from their legal representative, and were capable of being transported to Addenbrooke's Hospital. The exclusion criteria included any medical condition that made it unsafe for the patient to participate (decision made by clinical personnel blinded to the specific aims of the study) or any reason they are unsuitable to enter the MRI scanner environment (e.g. non-MRI-safe implants), significant pre-existing mental health problems, or insufficient English pre injury. After admission, each patient underwent clinical and neuroimaging testing. Patients spent a total of five days (including arrival and departure days) at Addenbrooke's Hospital. Coma recovery scale-revised (CRS-R) assessments were recorded at least daily for the five days of admission. If behaviours were indicative of awareness at any time, patients were classified as MCS; otherwise UWS. We assigned MCS− or MCS+ sub-classification if behaviours were consistent throughout the week. The most frequent signs of consciousness in MCS− patients are visual fixation and pursuit, automatic motor reactions (e.g. scratching, pulling the bed sheet) and localisation to noxious stimulation whereas MCS+ patients can, in addition, follow simple commands, intelligibly verbalise or intentionally communicate[58,65]. Scanning occurred at the Wolfson Brain Imaging Centre, Addenbrooke's Hospital, between January 2010 and December 2015; medication prescribed to each patient was maintained during scanning. All clinical investigations were conducted in accordance with the Declaration of Helsinki and all relevant ethical guidelines. Ethical approval for testing patients was provided by the National Research Ethics Service (National Health Service, UK; LREC reference 99/391).

As a focus of this study was on graph-theoretical properties of the brain, patients were systematically excluded from the final cohort analysed in this study

## Table 1 Demographic information for patients with Disorders of Consciousness

| Sex | Age | Months post injury | Aetiology | Diagnosis | CRS-R Score |
|---|---|---|---|---|---|
| M | 46 | 23 | TBI | UWS | 6 |
| M | 57 | 14 | TBI | MCS− | 12 |
| M | 46 | 4 | TBI | MCS | 10 |
| M | 35 | 34 | Anoxic | UWS | 8 |
| M | 17 | 17 | Anoxic | UWS | 8 |
| F | 31 | 9 | Anoxic | MCS− | 10 |
| F | 38 | 13 | TBI | MCS | 11 |
| M | 29 | 68 | TBI | MCS | 10 |
| M | 23 | 4 | TBI | MCS | 7 |
| F | 70 | 11 | Cerebral bleed | MCS | 9 |
| F | 30 | 6 | Anoxic | MCS− | 9 |
| F | 36 | 6 | Anoxic | UWS | 8 |
| M | 22 | 5 | Anoxic | UWS | 7 |
| M | 40 | 14 | Anoxic | UWS | 7 |
| F | 62 | 7 | Anoxic | UWS | 7 |
| M | 46 | 10 | Anoxic | UWS | 5 |
| M | 21 | 7 | TBI | MCS | 11 |
| M | 67 | 14 | TBI | MCS− | 11 |
| F | 55 | 6 | Hypoxia | UWS | 12 |
| M | 28 | 14 | TBI | MCS | 8 |
| M | 22 | 12 | TBI | MCS | 10 |
| F | 28 | 8 | ADEM | UWS | 6 |

CRS-R coma recovery scale-revised, UWS unresponsive wakefulness syndrome, MCS minimally conscious state, TBI traumatic brain injury

based on the following criteria: (1) large focal brain damage (i.e. more than 1/3 of one hemisphere) as stated by an expert in neuroanatomy blinded to the patients' diagnoses; (2) excessive head motion during resting state scanning (i.e. greater than 3 mm in translation and/or 3 degrees in rotation); (3) suboptimal segmentation and normalization of images. A total of 22 adults (14 males; 17–70 years; mean time post injury: 13 months) meeting diagnostic criteria for Unresponsive Wakefulness Syndrome/Vegetative State or Minimally Conscious State due to brain injury were included in this study (Table 1).

**Patient MRI data acquisition.** Resting-state fMRI was acquired for 10 min (300 volumes, TR = 2000 ms) using a Siemens Trio 3 T scanner (Erlangen, Germany). Functional images (32 slices) were acquired using an echo planar sequence, with the following parameters: $3 \times 3 \times 3.75$ mm resolution, TR = 2000 ms, TE = 30 ms, 78 degrees FA. Anatomical scanning was also performed, acquiring high-resolution T1-weighted images with an MPRAGE sequence, using the following parameters: TR = 2300 ms, TE = 2.47 ms, 150 slices, resolution $1 \times 1 \times 1$ mm.

**Preprocessing and denoising of patient fMRI data.** Due to the presence of deformations caused by brain injury, rather than relying on automated pipelines, patients' brains were individually preprocessed using SPM12, with visual inspections after each step. Additionally, to further reduce potential movement artefacts, data underwent despiking with a hyperbolic tangent squashing function. The remaining preprocessing and denoising steps were the same as described above for the propofol data.

**Intrinsic connectivity contrast.** The global connectivity of each voxel with the rest of the brain was analysed using the intrinsic connectivity contrast (ICC)[31] as implemented in the CONN toolbox. The ICC is a voxelwise measure of whole-brain connectivity based on the graph-theoretical notion of degree, which quantifies the number of connections of a node with other nodes in its network. Consistently with this notion, a higher ICC index corresponds to higher global connectivity.

For each voxel $i$,

$$\mathrm{ICC}(i) = \sum_j r\big((t(i), t(j))\big)^2 \tag{1}$$

where $i$, $j$ are voxels, and $t(x)$ is the BOLD fMRI timeseries of voxel $x$. Subsequently, the ICC values in the whole brain are normalised, by subtracting the average ICC value across all voxels from the ICC of each voxel, and then dividing each voxel's ICC by the standard deviation of the ICC values across all voxels.

The result is a brain map of voxelwise Intrinsic Connectivity Contrasts, whose distribution is Gaussian with zero mean and unitary variance. These ICC values

can then be used as ROIs for further seed-based analysis in a data-driven fashion. As the biological meaning of estimating functional connectivity of a white matter voxel from BOLD data is dubious, the ICC analysis was restricted to grey matter voxels.

**Voxelwise sample entropy of BOLD timeseries.** Sample entropy (SampEn) estimates the probability that similar sequences of observations in a timeseries will remain similar. To compute SampEn, the timeseries is divided into chunks of $m$ timepoints each; chunks are then compared to find the distance between them, calculated as the largest absolute difference between any value in the first chunk and any value in the second chunk (Chebyshev distance). Two chunks are deemed similar if their distance is less than a value $r$. Subsequently, the procedure is repeated for chunks of length $m + 1$. The result is the probability that if two data sequences of length $m$ have distance less than $r$ (i.e. are similar), then sequences of length $m + 1$ also have distance less than $r$. The negative logarithm of this quantity corresponds to SampEn.

$$\mathrm{SampEn} = -\log\frac{A}{B} \tag{2}$$

Here, $A$ is the number of chunks of length $m + 1$ that are similar (have Chebyshev distance less than $r$), and $B$ is the number of chunks of length $m$ that are similar.

Unlike Shannon entropy, SampEn depends on the choice of parameters $m$ and $r$. Here, we used $m = 3$ and $r = 0.6$ times the standard deviation of the data. These parameter values were identified as optimal for calculation of SampEn from resting-state fMRI data[32]. The entropy of the BOLD signal timeseries was calculated using the Brain Entropy Mapping Toolbox (BENtbx; https://cfn.upenn.edu/zewang/BENtbx.php)[32] implemented in MATLAB. Each subject's preprocessed and denoised resting-state fMRI timeseries were used as inputs for BENtbx. For each subject, the toolbox computed the SampEn of each voxel's timeseries, producing as output a subject-specific 3D brain map, with an entropy value in each voxel. Each subject-specific image was then spatially smoothed with the toolbox-recommended Gaussian kernel of 10 mm FWHM.

**Overlap of sample entropy and ICC results.** To identify the spatial overlaps between maps of statistical differences in ICC or SampEn values in DOC and anaesthesia, the fslmaths functions from FMRIB Software Library (FSL; https://fsl.fmrib.ox.ac.uk/fsl) was used to binarise the thresholded significance maps. For each dataset (anaesthesia and DOC), the binarised ICC and entropy masks were then superimposed on each other, to find regions that showed reductions in both entropy and integration. Thus, one mask of ICC-entropy overlap was obtained for each dataset. Finally, to identify whether the same regions showed common reductions of entropy and integration in both DOC and anaesthesia, the two dataset-specific masks of ICC-entropy overlap were themselves superimposed to find their overlap. The result was a single brain mask, showing those brain regions that significantly reduced both their temporal entropy and their global connectivity between conscious and unconscious conditions, both in the propofol and the DOC datasets.

**Connectivity matrix construction.** To construct matrices of functional connectivity, normalised brains were parcellated into 90 cortical and subcortical regions of interest (ROIs) derived from the automated anatomical labelling (AAL) atlas, covering the entire brain excluding Cerebellum and Vermis[66].

Each ROI was also assigned to one of 7 well-characterised resting-state networks (RSNs), derived from resting-state intrinsic connectivity analysis of 1000 healthy individuals[43]: default mode network (DMN), somatomotor (SOM), visual (VIS), salience/ventral attention network (SAL), dorsal attention network (DAN), fronto-parietal network (FPN), and limbic plus subcortical regions (LIM).

The timecourses of denoised BOLD signals were averaged between all voxels belonging to a given AAL-derived ROI, using the CONN toolbox. The resulting region-specific timecourses of each subject were then extracted for further analysis in MATLAB version 2016a. Functional connectivity was estimated as the Pearson correlation coefficient between the timecourses of each pair of ROIs, over the full scanning length.

**Dynamic functional connectivity.** Dynamic connectivity matrices were derived using an overlapping sliding-window approach[11,24]. For each subject and each condition, tapered sliding windows were obtained by convolving a rectangle of 22 TRs (44s) with a Gaussian kernel of 3 TRs, sliding with 1 TR step size. This resulted in 229 windows/timepoints per condition for the awake and anaesthetised subjects, and 273 windows for the DOC patients.

Within each of the resulting overlapping temporal windows of 22 TRs, a 90-by-90 matrix of functional connectivity was estimated, with the connection between each pair of AAL-derived ROIs being given by the Pearson correlation between their timecourses within that window.

**Derivation of integrated and segregated states.** States of higher integration or segregation were identified from the patterns of connectivity between regions, by establishing a "cartographic profile"[25,26] based on the module assignments of each ROI, considered as a network node. Firstly, within each time-resolved functional

connectivity matrix (weighted and signed), the asymmetric algorithm of Rubinov and Sporns[67] implemented in the MATLAB-based Brain Connectivity Toolbox (BCT; http://www.brain-connectivity-toolbox.net)[8] was used to identify network modules by applying the Louvain greedy algorithm[68], which iteratively evaluates different ways of assigning nodes to modules, in order to maximise the resulting modularity function $Q$:

$$Q = \frac{1}{v^+} \sum_j (w_{ij}^{+} - e_{ij}^{+}) \delta_{M_i M_j} - \frac{1}{v^+ + v^-} (w_{ij}^{-} - e_{ij}^{-}) \delta_{M_i M_j} \qquad (3)$$

where $v$ is the total weight of the graph (sum of all edges), $w_{ij}$ is the (signed) weight of the edge between nodes $i$ and $j$, $e_{ij}$ is the weight of an edge divided by the total weight of the graph (positive and negative edges are denoted with ' + ' and '−' superscripts, respectively), and $\delta_{MiMj}$ is set to 1 when nodes $i$ and $j$ are in the same module and 0 otherwise.

In the case of signed graphs, a module is defined as a group of nodes that are positively correlated with each other, but negatively correlated with nodes belonging to different modules[8]. Due to its stochastic nature, the algorithm was repeated for 100 iterations for each time-resolved network, and the module size resolution parameter $\gamma$ was set to one, the default[25,26].

Based on the modularity assignments identified in the previous step, we then derived the participation coefficient and within-degree Z-score for each node. The participation coefficient $P_i$ quantifies the degree of connection that a node entertains with nodes belonging to other modules: the more of a node's connections are towards other modules, the higher its participation coefficient will be ref. [8]. Conversely, the participation coefficient of a node will be zero if its connections

are all with nodes belonging to its own module.

$$P_i = 1 - \sum_{s=1}^{M} \left( \frac{\kappa_{is}}{k_i} \right)^2 \qquad (4)$$

Here, $\kappa_{is}$ is the strength of positive connections between node $i$ and other nodes in module $s$, $k_i$ is the strength of all its positive connections, and $M$ is the number of modules in the network, as identified by a given modularity detection algorithm. The participation coefficient ranges between zero (no connections with other modules) and one (equal connections to all other modules). A network with high average participation coefficient can be expected to have high levels of integration between its constituent modules.

Conversely, the within-module degree Z-score $Z_i$ is a measure of a node's connectivity with other nodes belonging to its module. It indicates how much larger (or smaller) the node's connections to other nodes in the module are, relative to the average connection strength within that module. A node with high within-module degree Z-score has stronger-than-average coupling with the other nodes in its module[8].

$$z_i = \frac{\kappa_{is} - \bar{\kappa}_{is}}{\sigma_{\kappa_{is}}} \qquad (5)$$

where $\kappa_{is}$ is the strength of connections between node $i$ and other nodes in module $s$, and $\bar{\kappa}_{is}$ and $\sigma_{\kappa is}$ are respectively the average and the standard deviation of $\kappa_{is}$ over all nodes belonging to module $s$. The Brain Connectivity Toolbox was used to derive the participation coefficient and within-degree Z-score for each node.

Subsequently, joint histograms of participation coefficient and within-module Z-score were produced for each timepoint (using MATLAB code made freely available by Shine et al. at https://github.com/macshine/integration/)[25], since together, these two measures quantify both a node's inter- modular and intra-modular connectivity. For each subject, the joint patterns were then used to assign each timepoint to one of two clusters, using an unsupervised machine learning algorithm known as k-means clustering (setting $k = 2$)[25]. To avoid the possibility of the algorithm becoming stuck in local minima, it was repeated 500 times with random re-initialisation of the two clusters' initial points. This was performed individually for each subject and condition. Following Shine et al.[25], Pearson correlation was chosen as distance metric for the algorithm.

Finally, the cluster with higher mean participation coefficient was labelled as the integrated state, while the cluster with lower average participation coefficient was considered to be the segregated state[25]. For each subject, a centroid matrix of functional connectivity was computed for each state, as the element-wise median of the timepoint-specific FC matrices assigned to the cluster corresponding to that state. The proportion of time spent in each state was also estimated, as the number of timepoints assigned to that cluster, over the total number of timepoints.

**Dynamic state connectivity analysis**. Once an integrated and a segregated state centroids had been derived for each subject and condition, we investigated how the pattern of connectivity in each state varied between the conscious and unconscious conditions. Since this work focuses on the common aspects of unconsciousness induced by different means, we sought to identify which connections were significantly affected in the same way in both datasets. This involved a two-step analysis. First, for each state (integrated, segregated) the corresponding centroid matrices were compared between Awake and Deep conditions, and between conscious controls and DOC patients. Thus, a matrix of mean differences was computed for each state in each dataset, so that the edge $W_{i,j}$ represented the mean

difference across conditions in the strength of the connection between regions $i$ and $j$. Subsequently, the matrices were significance-thresholded by setting to 0 any edge that did not reach a significance level of $\alpha < 0.05$ (FDR-corrected for multiple comparisons), as assessed by $t$-tests (paired for the propofol dataset, independent-samples for the DOC dataset). The resulting matrices of significant edges were then combined across datasets, to construct a matrix of common unconsciousness-induced connectivity differences. In this matrix, if both significance-thresholded matrices had a positive edge at position $i,j$, edge $w_{i,j}$ was set to the minimum value of the two; if both thresholded matrices had negative values at $i,j$, edge $w_{i,j}$ was instead set to the maximum of the two (i.e. smaller absolute value); and $w_{i,j}$ was set to 0 in all other cases. The presence of a non-zero edge in the resulting matrix therefore indicates that the corresponding regions $i$ and $j$ show consistent changes in their functional connectivity as a result of unconsciousness, regardless of how it was induced—with the direction of the alteration being given by the sign of $w_{i,j}$, and its weight representing the minimum extent of such alteration. After performing this analysis for the integrated and segregated state, the same analysis was also performed on the static FC matrices. This revealed whether any of the patterns of change observed in the static FC were specifically due to changes in the segregated or the integrated state.

**Brain graph construction**. In graph theory, a graph $G = (N,K)$, is a mathematical representation of a network of $N$ nodes (or vertices, typically represented as points) connected by $K$ edges (or links, typically represented as lines between pairs of points). Here, subject- and condition-specific brain graphs were constructed by thresholding the corresponding functional connectivity matrices (see above), so that nodes were given by AAL-derived brain regions, and edges were given by their functional connectivity. We thresholded each FC matrix proportionally, at density levels ranging between 10 and 25%, sampled in steps of 5%. To ensure robustness of the result, graph-theoretical metrics were averaged across thresholds for each subject and session before analysis.

Since edge weight can be expected to carry biological meaning as the strength of communication between different regions, we chose not to binarise the graphs. In line with similar work[9], we used weighted graphs for the analyses presented here, made possible by the availability of algorithms for weighted graph analysis[8].

To ensure that our results could not be attributed to our choice of node and edge estimation, we ran the same graph-theoretical analyses (for static FC, integrated state and segregated state, in both anaesthetised individuals and DOC patients) with different parameter choices: for edges, we applied higher density levels (30 to 50%, in 10% steps) and binarised rather than weighted edges; for nodes, we used the 234 ROIs of the Lausanne scale 125 template[69]) (Supplementary Figs. 11–14).

**Graph metrics**. The graph-theoretical properties used in this work are described below, based on the definitions provided by Rubinov and Sporns[8,67]. Each measure was computed for the static, integrated and segregated functional connectivity matrix of each subject, for each scanning session. All graph-theoretical measures were computed using the Brain Connectivity Toolbox[8].

Characteristic path length ($L$) is a network-wide measure of how effortful it is on average to move between different nodes in the network. This metric is calculated as the average length of the shortest path $d$ between every pair of nodes in the network.

$$L = \frac{1}{n} \sum_i^n \frac{\sum_{j \neq i}^n \left( d_{ij} \right)}{n - 1} \qquad (6)$$

The characteristic path length is understood as inversely related to the ability to integrate information across the whole network[8].

The clustering coefficient of node $i$ ($C_i$) is a node-specific measure of how well connected a node's neighbourhood is; in a binarized graph, it is calculated as the fraction of neighbours of the node that are also neighbours of each other.

$$C_i = \frac{2t_i}{k_i(k_i - 1)} \qquad (7)$$

where $t_i$ is the number of triangles around node $i$, (for binarized graphs) or the geometric mean of triangles around node $i$ (for weighted graphs). The mean of all nodes' clustering coefficients (i.e. the network's mean clustering coefficient) indicates how well connected, on average, the neighbourhoods present in the network tend to be. When applied to brain networks, the clustering coefficient is thought to represent the degree of information integration at a local level, and hence the potential for efficiently performing specialised, segregated local processing[8].

Both the characteristic path length and mean clustering coefficient are typically provided as normalised, i.e. divided by the corresponding metric of a random graph with the same number of nodes and edges, and preserved degree distribution[8]. Here, we used the average of the same measures computed for 100 random graphs, each constructed by randomly rewiring each edge in the original graph 100 times.

Small-worldness: a network is considered to be small-world if it has both the high mean clustering coefficient typical of regular lattice networks, but also the small characteristic path length typical of random networks[40]. Small-worldness

represents the balance between integration/global processing (low characteristic path length) and segregation/local processing (high mean clustering coefficient), and is summarised by S, the ratio of the graph's normalised mean clustering coefficient to normalised characteristic path length[8]..

$$S_{graph} = \frac{\frac{C_{graph}}{C_{random}}}{\frac{L_{graph}}{L_{random}}} \qquad (8)$$

**Connectivity entropy**. Complexity of a given state of functional connectivity was estimated by the mean normalised Shannon entropy of the connections of each ROI. Following the procedure of Saenger et al.[42], the connectivity values in each column of the matrix were assigned into $n$ bins to construct a distribution, and a normalised version of the Shannon entropy of this distribution was computed for each node according to the following equation[42]:

$$H = -\sum_{i=1}^{n} p_i \log(p_i)/\log(n) \qquad (9)$$

To constrain the entropy values between 0 and 1[42], the original Shannon entropy was normalised by the Shannon entropy of a uniform distribution, which corresponds to $\log(n)$. Here, we followed Saenger and colleagues[42], always using $n = 10$ bins.

**Statistical analysis**. The statistical significance of within-group differences between the awake and deep anaesthetised conditions was determined with non-parametric permutation $t$-tests (repeated-measures), with 10,000 permutations. Permutation- based two-samples t-tests with 10,000 permutations were instead used to assess group differences between the awake condition of the propofol dataset (used as healthy control group) and the DOC patients, and between the TBI and HBI subgroups of DOC patients. All tests were two-sided. For measures that showed a significant difference, the effect size was also estimated using Hedge's g.

When subnetwork-specific analyses were performed, the Benjamini–Hochberg procedure[70] was adopted to control the false discovery rate across multiple comparisons, at a corrected $\alpha$ level of 0.05. This procedure was also applied for the analysis of FC matrix differences.

For the statistical analyses performed on brain maps (ICC and SampEn), group-level comparisons were implemented as a GLM, using parametric paired t-tests for the differences between awake and deep anaesthesia conditions, and unpaired $t$-tests for the differences between DOC patients and controls. All tests were two-sided. The resulting output maps of group differences were thresholded at voxelwise $p < 0.001$ (uncorrected), and corrected for multiple comparisons by applying a family-wise error (FWE) cluster-based correction, resulting in $p < 0.05$. These steps were performed within the SPM12-based CONN toolbox for the ICC analysis, and directly in SPM12 for the SampEn analysis.

**Reporting summary**. Further information on research design is available in the Nature Research Reporting Summary linked to this article.

## Data availability
The data that support the findings of this study are available from the corresponding author upon reasonable request. Source data underlying Figs. 2 and 3, Supplementary Figs. 10–14 and Supplementary Tables 6–17 are provided as a Source Data file.

## Code availability
The CONN toolbox is freely available online (http://www.nitrc.org/projects/conn). Code for the "cartographic profile"[25] is freely available online (https://github.com/macshine/integration/). The Brain Connectivity Toolbox code used for graph-theoretical analyses is freely available online (https://sites.google.com/site/bctnet/). The Brain Entropy Toolbox is freely available online (https://cfn.upenn.edu/zewang/BENtbx.php). The code used to compute the Sample Entropy (SampEn) of motion timeseries in MATLAB is freely available online (https://uk.mathworks.com/matlabcentral/fileexchange/35784-sample-entropy).

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

## Acknowledgements

This work was supported by grants from the UK Medical Research Council (U.1055.01.002.00001.01) [to A.M.O. and J.D.P.]; The James S. McDonnell Foundation [to A.M.O. and J.D.P.]; The Canada Excellence Research Chairs program (215063) [to A.M.O.]; The Canadian Institute for Advanced Research (CIFAR) [to A.M.O., D.K.M. and E.A.S.]; The National Institute for Health Research (NIHR, UK), Cambridge Biomedical Research Centre and NIHR Senior Investigator Awards [to J.D.P. and D.K.M.]; The British Oxygen Professorship of the Royal College of Anaesthetists [to D.K.M.]; The Cambridge International Trust and the Howard Sidney Sussex Studentship [to M.M.C.]; The Oon Khye Beng Ch'Hia Tsio Studentship for Research in Preventive Medicine, Downing College, University of Cambridge [to I.P.]; The Evelyn Trust, Cambridge and the EoE CLAHRC fellowship [to J.A.]; The L'Oreal-Unesco for Women in Science Excellence Research Fellowship [to L.N.]; The Stephen Erskine Fellowship, Queens' College, University of Cambridge [to E.A.S.] and the Gates Cambridge Trust [to A.I.L.]. The research was also supported by the NIHR Brain Injury Healthcare Technology Co-operative based at Cambridge University Hospitals NHS Foundation Trust and University of Cambridge. Computing infrastructure at the Wolfson Brain Imaging Centre (WBIC-HPHI) was funded by the MRC research infrastructure award (MR/M009041/1). We would like to thank Victoria Lupson and the staff in the Wolfson Brain Imaging Centre (WBIC) at Addenbrooke's Hospital for their assistance in scanning. We would also like to thank Dian Lu and Thomas Varley for useful discussions, and all the participants for their contribution to this study.

## Author contributions

A.I.L. and E.A.S. designed and carried out this study. A.I.L. analysed the data. M.M.C., I.P. and E.A.S. contributed to data analysis. P.F., G.B.W., J.A., J.D.P., A.M.O., L.N., D.K.M. and E.A.S. were involved in designing the original studies for which the present data were collected. P.F., M.M.C., G.B.W., J.A., L.N. and E.A.S. all participated in data collection. A.I.L., D.K.M. and E.A.S. wrote the paper with feedback from all co-authors.

## Competing interests

The authors declare no competing interests.
