## [Peer Review File · Nature Communications]

Reviewers' comments:

Reviewer #1 (Remarks to the Author):

The article by Luppi and colleagues presents an interesting evaluation of different metrics associated with loss of consciousness (entropy, integration, and their dynamic counterparts). Their manuscript is attractive in many ways, I consider it especially solid that the authors decided to compare data from two groups with unconsciousness elicited by brain injury and by general anesthesia.

The results are very interesting and complement well recent articles following a similar approach. Of these, I consider the authors should cite and discuss the following work, which presents a substantial overlap with their own:

Demertzi, A., Tagliazucchi, E., Dehaene, S., Deco, G., Barttfeld, P., Raimondo, F., ... & Schiff, N. D. (2019). Human consciousness is supported by dynamic complex patterns of brain signal coordination. *Science advances*, 5(2), eaat7603.

Should the authors identify other recent articles studying unconsciousness using similar methods, I believe they should be incorporated to the discussion too.

I have some comments aiming to improve the article before publication:

1. The authors performed many analyses which are well described in the Methods section. Because of this, the section is very long. Reading the manuscript in sequential order can be tedious because of this. An alternative would be to introduce the methodological concepts along with the results, and leave the Methods as the final section of the manuscript. Of course, this is only a suggestion and ultimately a decision to be shared between the authors and the editors of the manuscript.
2. It seems the authors pooled together UWS and MCS patients, but I think they should consider whether this is valid since both groups are expected to have different levels of awareness.
3. Also the authors pooled brain injured patients with very different etiologies; they should consider evaluating or further discussing the heterogeneity of the sample.
4. Both under deep sedation and in brain injured patients, head movement can be an issue. This is especially true in the case of signal entropy. The variance of the head motion time series is not the adequate variable to compare between groups to assess the potential confound of movement in entropy, because even a small amplitude but high entropy head movement could bias the entropy estimates. Even if the subjects are quieter (e.g. deep sedation), the entropy of the signals can be higher - and most likely it will be, since awake subjects will move in ways presenting long-range correlations, while entropy for a completely still subject will most likely reflect that of white noise associated with the measurement itself. In short, it would be interesting to see a comparison of these metrics between head movement time courses of the different groups and conditions.
5. The participation coefficient is not really a measure of diversity, but of the degree of connection of a node with other modules. The links could be all towards the same module and, if the number is high enough, this could lead to a high participation coefficient without diversity in the involved modules.
6. I have spotted a couple of LaTeX compilation errors, please double check.

Reviewer #2 (Remarks to the Author):

Luppi and colleagues presented a very timely research, looking at the common neural signature of reduction of differentiation and integration due to loss of consciousness due to general anesthesia and brain injury based on fMRI data.

The manuscript is very clearly written. The data are exhaustively analyzed and clearly presented. The controversial viewpoints are represented in a balanced way. The results are novel. Thus I don't have much comments to say. I list minor comments below.

The only request, if I may, is to make the data and the codes publicly available in some way to promote Open Science. If opening the raw data is difficult due to the patients' ID issues, then that is fine. Maybe some processed data can be still made available?

In any case, this is a great paper. Congratulations!

Minor

L73 : Reference to Ramsay score?

L76 : explanation of the two tasks? Are they the valid tasks to assess level of wakefulness? The former could be performed if auditory processing is locally lost. The latter could be performed less if memory is impaired.

L107 : "Since the latter did not take place" -> Unclear what "the latter" refers to from the context.

L147 "The aCompCor" Typo?

L231 Equation 2. Inside of $U()$ has $| |$, which usually means absolute value. This is a bit confusing in this context. Better removed.

L359 therefore that -> therefore "indicate?" that

Equation (6) missing ")"

Most equations with inequality signs are not displayed correctly.

Ref 19 : Author order is wrong. Also, a recent paper by Casarrotto et al 2016 "Stratification of Unresponsive Patients by an Independently Validated Index of Brain Complexity" is highly relevant to this paper, and better to be mentioned somewhere in this paper.

Sup Fig 3 caption "m=0.3" -> m=3 ?

Sup Fig 4 and 5. Are the legends correct? There are no explanation of each panel a)-f) but it points to left/right. (Though I can more or less understand what they are showing...)

Signed review by Naotsugu Tsuchiya, Monash University

Reviewer #3 (Remarks to the Author):

In this prospective study of 16 healthy adults undergoing anesthesia and 22 patients with disorders of consciousness (DoC) caused by severe brain injuries, Luppi and colleagues analyze resting-state functional MRI (rs-fMRI) data to investigate functional network properties that distinguish consciousness from unconsciousness. Multiple quantitative measures of functional connectivity are derived from the rs-fMRI data, including graph theoretical measures and measures of entropy. These quantitative measures are selected based on prior literature and on conceptual models of consciousness postulating that these measures reflect network properties

essential for consciousness (as predicted by Integration Information Theory and Global Neuronal Workspace Theory).

There are several key findings, which collectively provide new insights into the functional correlates of consciousness in the resting human brain. Most notably, the authors show that unconsciousness is associated with decreased functional diversity and integrative capacity in the posterior cingulate node of the default mode network (DMN). This finding is consistent with extensive prior evidence for the key role of the posterior cingulate as a hub, or “hot spot”, for conscious processing. This finding addresses a cited discrepancy in the literature (Ref 14) and expands current knowledge about the spatial and temporal dynamics of the posterior cingulate in conscious processing. Another new insight is that cortico-cortical and thalamo-cortical connections within the DMN have different impacts on consciousness depending on the brain's state of integration or segregation. Overall these observations add new detail to a rapidly emerging field that seeks to map the neural correlates of human consciousness.

Overall, the paper is well written and thoroughly cited. The methods and statistical analyses appear rigorous. Testing functional biomarkers in two types of unconsciousness (anesthesia and severe brain injury) is a particularly elegant approach that highlights the potentially fundamental nature of the insights revealed here, even if the study appears to be retrospective rather than prospectively designed. There are additional examples of methodological rigor, including assessment of the Ramsay level by multiple clinicians, performance of auditory and cognitive tests to confirm the state of unconsciousness in the anesthesia cohort, presence of an infrared camera inside the scanner to assess for wakefulness, testing the functional connectivity matrix at different density thresholds, as well as repeating the analysis using different node delineation strategies and different edge measures (i.e. both weighted and binarized).

The manuscript can be improved by addressing the following issues:

Major Comments:

1. The authors state that inclusion of a brain injury cohort allows them to avoid the potential confound that many causes of unconsciousness “...have widespread effects on the brain, affecting multiple brain functions in addition to those specifically relevant for consciousness” (line 47). However, 10 of the 22 patients with DoC experienced a hypoxic-ischemic brain injury. This type of injury, as the authors know, is often global with respect to cortical, basal ganglionic and diencephalic involvement (although the brainstem may be spared). Does the presence of global or near-global injury in these patients undermine the authors' claim that they have mitigated the potential confound described above? The authors should consider adding separate analyses for TBI and HIE in the supplement.
2. In Table 1, please clarify why some patients are categorized as MCS- vs. MCS+, whereas some are categorized as MCS.
3. The control for the DOC group is not clear. Is the pre-propofol data in healthy subjects obtained using a different scanner and scanning protocol being used as the control for the brain injury cohort? If so, the comparisons are not really independent and some discussion of the implications of both using different acquisitions for control and brain injury and a common control group should be added to the Discussion. The Methods should also be clearer with regard to the control group for the brain injury analyses.
4. The mention of a 5-minute auditory story task (bottom of page 5) is confusing. Does that mean that the propofol subjects listened to a story for 5 of the 8 minutes of scanning? Presumably that was for another purpose like evaluating LOC effects on language networks. Or is that a completely different acquisition that was not analyzed? This could be clearer. If the propofol data analyzed in this work was partially acquired during the story task, this could be a confound since propofol anesthesia was light and a story task is an alerting stimulus which could cause task-dependent DMN suppression?
5. Is there any concern that spontaneously ventilating subjects on propofol are hypoventilating and hypercarbic, potentially altering BOLD signals?

6. While several limitations are addressed in the discussion, more attention and emphasis should be placed on potential limitations of the graph theoretical measures that are utilized here. These limitations are discussed in several recent reviews, such as in Papo et al. Beware of the Small-World Neuroscientist. *Frontiers in Human Neuroscience*, 2016.

Minor Comments:

1. There are several issues with the references:
 - a. Reference 7 is missing the year, volume, and page numbers.
 - b. Reference 9, 14, 15, and 30 are missing the year
 - c. Reference 19 has an incorrect author list – the first author of this paper is Casali, not Massimini
2. (line 1) “The brain is a remarkably complex system, occupying a near-critical point...” here “occupying” seems to be a poor choice of verbs because it anthropomorphizes and because the brain as a whole is not really near a critical point but maybe its activity patterns are? Not 100% sure if that is a valid criticism or reflection of the criticality notion.
3. Line 22 – to avoid inadvertently giving the impression that resting EEG can provide a measurement of the Perturbational Complexity Index, the authors should clarify that these EEG measurements are made after a TMS pulse.
4. For the sentence that references the *Sci Trans Med* 2013 paper (ref 19), would recommend citing follow-up papers by Massimini’s lab that provide further evidence of the use of the Perturbation Complexity Index as a measure of consciousness: Casarotto et al. *Annals of Neurology* 2016 and Comolatti et al. *Brain Stimulation* 2019.
5. The recent paper by Demertzi et al published in *Science Advances* is quite relevant to the present paper and should be cited.
6. (line 83) Baxter A50 (Singapore) should be explained.
7. Was a 32-channel coil used for both the rs-fMRI and the T1 MPRAGE sequence? The 32-channel coil is only mentioned in relation to the T1 MPRAGE sequence.
8. The use of anisotropic voxels for the patient T1 sequence (1 x 1 x 1.2 mm) is another minor weakness.
9. Table 1 – I assume the authors performed the CRS-R not the CRS – this should be clarified.
10. Line 359 – the sentence appears to be missing a verb.
11. There are some redundancies, with repetition of methods in the results section (e.g. lines 464 to 471, and lines 514-526).
12. There are also interpretation statements in the results that are more appropriate for the discussion (e.g. line 476-477).
13. For the sentence on 510, in addition to references 48-53, consider citing Vanhaudenhuyse et al *Brain* 2010 and Threlkeld et al *Cortex* 2018; also consider citing Threlkeld et al on line 675.
14. (lines 591 and 596) Is it correct to say that the global connectivity entropy was “driven” by the changes in the DMN and other networks? Although the mathematical result may have primarily reflected DMN effects, weren’t these ultimately just correlated? “Driven” implies a causal relationship such that reduced connectivity entropy in the DMN would cause global connectivity entropy changes, which was not proven in these experiments.
15. Some of the text in the Supplementary Material is redundant with the main text.
16. The layout of supplementary Figures 4-6 is not as intended and does not match their legend.

Authors' Response to Reviewer Comments

First of all, we would like to thank the Editor for considering our manuscript, and all three Reviewers for their insightful and constructive feedback. In this revised version of the manuscript, we did our best to address all comments raised by the Reviewers.

Specifically, (i) we added additional analyses of the entropy of motion parameters; (ii) we added analyses to compare the subgroups of patients with Disorders of Consciousness due to hypoxic/ischemic brain injury and traumatic brain injury; (iii) we re-structured the organisation of the manuscript, as suggested, to improve readability; (iv) we added discussions of the limitations of using graph-theoretical measures such as small-worldness, of the possibility of hypercapnia in the anaesthetised volunteers, and of the use of behavioural unresponsiveness as a marker of unconsciousness; (v) we added missing references, either suggested by the Reviewers or appeared during the period the paper was under review; (vi) we corrected typos and errors across the document.

A detailed item-by-item response to each of the Reviewers' points follows, with our replies marked in blue.

Point-to-point response to Reviewers' comments:

Reviewer #1 (Remarks to the Author):

The article by Luppi and colleagues presents an interesting evaluation of different metrics associated with loss of consciousness (entropy, integration, and their dynamic counterparts). Their manuscript is attractive in many ways, I consider it especially solid that the authors decided to compare data from two groups with unconsciousness elicited by brain injury and by general anesthesia.

The results are very interesting and complement well recent articles following a similar approach. Of these, I consider the authors should cite and discuss the following work, which presents a substantial overlap with their own:

Demertzi, A., Tagliazucchi, E., Dehaene, S., Deco, G., Barttfeld, P., Raimondo, F., ... & Schiff, N. D. (2019). Human consciousness is supported by dynamic complex patterns of brain signal coordination. *Science advances*, 5(2), eaat7603.

Should the authors identify other recent articles studying unconsciousness using similar methods, I believe they should be incorporated to the discussion too.

I have some comments aiming to improve the article before publication:

We thank the reviewer for their constructive comments which we address in detail below.

1. The authors performed many analyses which are well described in the Methods section. Because of this, the section is very long. Reading the manuscript in sequential order can be tedious because of this. An alternative would be to introduce the methodological concepts along with the results, and leave the Methods as the final section of the manuscript. Of course, this is only a suggestion and ultimately a decision to be shared between the authors and the editors of the manuscript.

We are grateful to the reviewer for providing this suggestion on how to make our work easier to read. We agree that the proposed format would increase the legibility of the manuscript, and we have now re-written the manuscript to follow this suggestion. This reformatting adheres to the format requirements for Nature Communications.

2. It seems the authors pooled together UWS and MCS patients, but I think they should consider whether this is valid since both groups are expected to have different levels of awareness.

The reviewer raises an important point that we believe deserves further discussion. To this end, we have added the following paragraph to the Discussion, to reflect this concern and broader issues with using behavioural unresponsiveness as a proxy for unconsciousness:

[...] in this work we have implicitly assumed that both individuals under deep propofol anaesthesia and UWS and MCS patients are unconscious. However, MCS patients owe their classification to occasional signs of volitional behaviour, which may reflect minimal levels of consciousness⁵⁸. Furthermore, disorders of consciousness are prone to relatively high rate of misdiagnosis, with patients categorized as UWS subsequently exhibiting signs of awareness when more sensitive measures are employed⁵⁷. Adding to this complication, dreaming has been reported during anaesthesia in up to 27% of cases⁵⁹. Consequently, like in most studies of this kind, despite lack of behavioural responsiveness it is not possible to say conclusively that all individuals examined here were completely unconscious, in the sense of having no subjective experiences. Using additional markers of consciousness, such as SWAS⁴⁶, PCI¹⁹ or naturalistic paradigms⁵⁷ may be required in future studies to provide additional evidence of unconsciousness independent of behavioural responsiveness.

3. Also the authors pooled brain injured patients with very different etiologies; they should consider evaluating or further discussing the heterogeneity of the sample.

To reassure the reviewer and ourselves that our patient groups were broadly comparable, we have now added a separate analysis to compare patients with TBI and patients with hypoxic/ischemic injury (Results, Supplementary Table 17 and Supplementary Figure 13). The analysis (using permutation-based two-samples t-tests) did not reveal any significant differences between the two groups of patients (although we acknowledge that the limited size of each group of patients may have limited our ability to detect subtler differences).

4. Both under deep sedation and in brain injured patients, head movement can be an issue. This is especially true in the case of signal entropy. The variance of the head motion time series is not the adequate variable to compare between groups to assess the potential confound of movement in entropy, because even a small amplitude but high entropy head movement could bias the entropy estimates. Even if the subjects are quieter (e.g. deep sedation), the entropy of the signals can be higher - and most likely it will be, since awake subjects will move in ways presenting long-range correlations, while entropy for a completely still subject will most likely reflect that of white noise associated with the measurement itself. In short, it would be interesting to see a comparison of these metrics between head movement time courses of the different groups and conditions.

Following this comment, we performed statistical analyses to compare the Sample Entropy of each of the six head motion parameter time courses (three translations and three rotations) between the awake healthy controls, the same volunteers under anaesthesia, and the DOC patients (please see article text below).

“We also compared the Sample Entropy of the head motion signals (three translations and three rotations) between the awake healthy controls and the two conditions of unconsciousness, using the same parameters as for the brain entropy analysis described above; although a significant reduction in the entropy of the head motion in the horizontal plane (x-translation) was observed when comparing DOC patients to controls (Supplementary Table 6), this was not the case for deep propofol anaesthesia, where instead an increase in the entropy of the timeseries of rotations around the vertical axis (z-rotation) was observed, compared with the awake condition (Supplementary Table 7).”

Since our brain signal results indicate reductions in entropy in both anaesthesia and DoC, we conclude that the entropy findings presented in the manuscript are not merely due to differences in subject motion, since they are common to both states of unconsciousness, unlike the differences in

subject motion entropy, which pertain to different motion parameters (x-translation and z-rotation) and go in opposite directions. Finally, this concern is further mitigated by the denoising procedures that we adopted, which included regressing out the subject motion parameters and their first derivatives.

5. The participation coefficient is not really a measure of diversity, but of the degree of connection of a node with other modules. The links could be all towards the same module and, if the number is high enough, this could lead to a high participation coefficient without diversity in the involved modules.

We thank the reviewer for this comment; the description of the participation coefficient in the Methods section of the manuscript has been re-formulated using the more accurate phrasing proposed by the reviewer. Please see below:

“The participation coefficient P_i quantifies the degree of connection that a node entertains with nodes belonging to other modules: the more of a node’s connections are towards other modules, the higher its participation coefficient will be⁸. Conversely, the participation coefficient of a node will be zero if its connections are all with nodes belonging to its own module.”

6. I have spotted a couple of LaTeX compilation errors, please double check.

We thank the reviewer for bringing these errors to our attention; we have proofread the manuscript again following re-writing in Word and we have rectified these errors.

Reviewer #2 (Remarks to the Author):

Luppi and colleagues presented a very timely research, looking at the common neural signature of reduction of differentiation and integration due to loss of consciousness due to general anesthesia and brain injury based on fMRI data.

The manuscript is very clearly written. The data are exhaustively analyzed and clearly presented. The controversial viewpoints are represented in a balanced way. The results are novel. Thus I don't have much comments to say. I list minor comments below.

The only request, if I may, is to make the data and the codes publicly available in some way to promote Open Science. If opening the raw data is difficult due to the patients' ID issues, then that is fine. Maybe some processed data can be still made available?

In any case, this is a great paper. Congratulations!

We are very grateful to Professor Naotsugu Tsuchiya for his encouraging comments. We address his specific comments below.

Minor

L73 : Reference to Ramsay score?

We have added a reference to the original Ramsay article.

L76 : explanation of the two tasks? Are they the valid tasks to assess level of wakefulness? The former could be performed if auditory processing is locally lost. The latter could be performed less if memory is impaired.

The concern that each task individually may be insufficient to assess level of wakefulness, as pointed out by the reviewer, is well received. To clarify our experimental procedure, data collection only began

when participants stopped responding to *both* computerised tasks, and the three independent assessors agreed that the desired level of anaesthetic depth had been reached. Given these conditions we are confident that the level of wakefulness was assessed comprehensively.

L107 : "Since the latter did not take place" -> Unclear what "the latter" refers to from the context.

We apologise for the lack of clarity, and we thank the reviewer for pointing it out to us. We have now clarified in the manuscript that "the latter" refers to the sedation procedure

L147 "The aCompCor" Typo?

We have clarified in the manuscript that this refers to the anatomical CompCor method of denoising the fMRI data and provided a reference.

L231 Equation 2. Inside of $U(\)$ has $| |$, which usually means absolute value. This is a bit confusing in this context. Better removed.

We thank the reviewer for this comment, we noticed that our description of the ICC in the Methods section was inaccurately stating that only positive connections would be used to compute the ICC. We have now amended the definition and simplified the corresponding equation in the manuscript to make it clearer:

"We estimated this [global integrative capacity] with the Intrinsic Connectivity Contrast (ICC), a voxel-wise measure that uses graph theory to quantify each voxel's whole-brain connectivity, computed as the sum of the squared Pearson correlation between that voxel's timeseries and the timeseries of every other voxel in the brain²⁸ "

L359 therefore that -> therefore "indicate?" that

Indeed, "indicate" was the missing verb, and it has now been corrected in the manuscript.

Equation (6) missing ")"

Most equations with inequality signs are not displayed correctly.

We have now re-written all equations to ensure legibility. We have used Word format for the revised version of the manuscript.

Ref 19 : Author order is wrong. Also, a recent paper by Casarrotto et al 2016 "Stratification of Unresponsive Patients by an Independently Validated Index of Brain Complexity" is highly relevant to this paper, and better to be mentioned somewhere in this paper.

We thank the reviewer for pointing out this inaccuracy which has been rectified in the revised version of the manuscript. The author order has been fixed, and the mentioned paper cited, as well as other recent work from the same group.

Sup Fig 3 caption "m=0.3" -> m=3 ?

The typo has been corrected in the figure caption.

Sup Fig 4 and 5. Are the legends correct? There are no explanation of each panel a)-f) but it points to left/right. (Though I can more or less understand what they are showing...)

We apologise for the discrepancy between figures and legends. The seed-based correlation results for DoC and propofol anaesthesia have now been divided into separate figures for simplicity, and the captions have been amended accordingly.

Signed review by Naotsugu Tsuchiya, Monash University

Reviewer #3 (Remarks to the Author):

In this prospective study of 16 healthy adults undergoing anesthesia and 22 patients with disorders of consciousness (DoC) caused by severe brain injuries, Luppi and colleagues analyze resting-state functional MRI (rs-fMRI) data to investigate functional network properties that distinguish consciousness from unconsciousness. Multiple quantitative measures of functional connectivity are derived from the rs-fMRI data, including graph theoretical measures and measures of entropy. These quantitative measures are selected based on prior literature and on conceptual models of consciousness postulating that these measures reflect network properties essential for consciousness (as predicted by Integration Information Theory and Global Neuronal Workspace Theory).

There are several key findings, which collectively provide new insights into the functional correlates of consciousness in the resting human brain. Most notably, the authors show that unconsciousness is associated with decreased functional diversity and integrative capacity in the posterior cingulate node of the default mode network (DMN). This finding is consistent with extensive prior evidence for the key role of the posterior cingulate as a hub, or "hot spot", for conscious processing. This finding addresses a cited discrepancy in the literature (Ref 14) and expands current knowledge about the spatial and temporal dynamics of the posterior cingulate in conscious processing. Another new insight is that cortico-cortical and thalamo-cortical connections within the DMN have different impacts on consciousness depending on the brain's state of integration or segregation. Overall these observations add new detail to a rapidly emerging field that seeks to map the neural correlates of human consciousness.

Overall, the paper is well written and thoroughly cited. The methods and statistical analyses appear rigorous. Testing functional biomarkers in two types of unconsciousness (anesthesia and severe brain injury) is a particularly elegant approach that highlights the potentially fundamental nature of the insights revealed here, even if the study appears to be retrospective rather than prospectively designed. There are additional examples of methodological rigor, including assessment of the Ramsay level by multiple clinicians, performance of auditory and cognitive tests to confirm the state of unconsciousness in the anesthesia cohort, presence of an infrared camera inside the scanner to assess for wakefulness, testing the functional connectivity matrix at different density thresholds, as well as repeating the analysis using different node delineation strategies and different edge measures (i.e. both weighted and binarized).

The manuscript can be improved by addressing the following issues:

Major Comments:

1. The authors state that inclusion of a brain injury cohort allows them to avoid the potential confound that many causes of unconsciousness "...have widespread effects on the brain, affecting multiple brain functions in addition to those specifically relevant for consciousness" (line 47). However, 10 of the 22 patients with DoC experienced a hypoxic-ischemic brain injury. This type of injury, as the authors know, is often global with respect to cortical, basal ganglionic and diencephalic involvement (although the brainstem may be spared). Does the presence of global or near-global injury in these

patients undermine the authors' claim that they have mitigated the potential confound described above? The authors should consider adding separate analyses for TBI and HIE in the supplement.

We agree with the reviewer that it is important to consider the different aetiologies present in our sample, and following the reviewer's suggestion, we repeated our analyses to compare the two DOC aetiologies separately thus ensuring that the two groups of patients were broadly comparable (Results, Supplementary Table 17 and Supplementary Figure 13). The analysis did not reveal any significant differences between patients with traumatic brain injury and hypoxic brain injury. We have also reformulated the statement in question in the revised manuscript as shown below:

"A further consideration is that there are multiple ways in which loss of consciousness may occur, whether through pharmacological interventions having widespread effects on brain function, or hypoxic-ischemic injuries affecting cortical and subcortical regions of the brain, or relatively localised traumatic brain injuries. In order to identify neurobiological signatures of loss of consciousness that are generalisable across these multiple conditions, rather than being specific to any of them, here we investigate alterations in brain functions and dynamics during unconsciousness arising in common from all of the above-mentioned causes."

2. In Table 1, please clarify why some patients are categorized as MCS- vs. MCS+, whereas some are categorized as MCS.

We have updated the corresponding section in the Methods, adding the statement below to clarify the diagnostic criteria.

Patients spent a total of five days (including arrival and departure days) at Addenbrooke's Hospital. Coma Recovery Scale-Revised (CRS-R) assessments were recorded at least daily for the five days of admission. If behaviours were indicative of awareness at any time, patients were classified as MCS; otherwise UWS. We assigned MCS- or MCS+ sub-classification if behaviours were consistent throughout the week. The most frequent signs of consciousness in MCS- patients are visual fixation and pursuit, automatic motor reactions (e.g. scratching, pulling the bed sheet) and localisation to noxious stimulation whereas MCS+ patients can, in addition, follow simple commands, intelligibly verbalise or intentionally communicate^{58,65}.

3. The control for the DOC group is not clear. Is the pre-propofol data in healthy subjects obtained using a different scanner and scanning protocol being used as the control for the brain injury cohort? If so, the comparisons are not really independent and some discussion of the implications of both using different acquisitions for control and brain injury and a common control group should be added to the Discussion. The Methods should also be clearer with regard to the control group for the brain injury analyses.

We thank the reviewer for this comment. Indeed, the reviewer's understanding is correct: in the present work, the healthy volunteers during wakefulness constituted the control group for both deep anaesthesia and the DOC patients. We have made this more explicit in the revised manuscript both in the introduction and methods sections, and the following paragraphs have been added to the discussion, to address the points raised by the reviewer:

"The present study also has a number of other limitations. Firstly, the scanners and acquisition parameters were not identical for the two cohorts discussed in the article. A further confound may be that DOC patients had reduced entropy of motion timeseries in the horizontal plane (x-axis translation) compared to awake controls, whereas anaesthesia led to increased entropy in the vertical rotation (z-axis). Additionally, it is important to consider how the anaesthetic agent can indirectly affect measures of brain activity by altering physiological parameters, such as arterial concentration of carbon dioxide. Since anaesthetized subjects in this study were not intubated but

rather spontaneously ventilating, propofol-induced respiratory depression may have resulted in hypoventilation and increased arterial CO₂ levels. We did not measure end-tidal or arterial CO₂ concentrations in our subjects; however, propofol only has relatively minor effects on brain hemodynamics⁶⁰, and BOLD fluctuation amplitudes, connectivity strength and the spatial extent of connectivity maps have been shown to be unaffected by hypercapnia during the resting state in rats⁶¹, thus mitigating this concern.

In addition to noting that both motion and cardiac, respiratory and physiological noise artifacts are accounted for in our denoising procedure (see Methods), we believe that these concerns should be further mitigated by our decision to adopt a comparative approach in this work, focusing only on results that were observed in both DOC and anaesthesia: if any of the results we observed when comparing DOC patients and controls had been due solely to the differences in acquisition and scanning parameters, or subject motion, such results should not have also been observed under conditions of anaesthesia, where those confounds were not present thanks to the within-subjects design— and vice versa for the concern about hypercapnia during anaesthesia. Moreover, using the same group of awake healthy volunteers as controls for both states of unconsciousness helps to ensure that the results we have found are not confounded by differences in control groups, but rather represent deviations from the same common baseline. Therefore, we expect that the common results reported here should represent genuine markers of loss of consciousness, rather than reflecting any idiosyncratic aspects of the specific datasets used here.”

4. The mention of a 5-minute auditory story task (bottom of page 5) is confusing. Does that mean that the propofol subjects listened to a story for 5 of the 8 minutes of scanning? Presumably that was for another purpose like evaluating LOC effects on language networks. Or is that a completely different acquisition that was not analyzed? This could be clearer. If the propofol data analyzed in this work was partially acquired during the story task, this could be a confound since propofol anesthesia was light and a story task is an alerting stimulus which could cause task-dependent DMN suppression?

We acknowledge the lack of clarity on this point in the previous version of the manuscript, and we have re-phrased the section in the revised version of the manuscript to explain that the 5-minute story scan is separate from the 8-minute resting state scan analysed in the present work (see paragraph below).

“In the scanner, subjects were instructed to relax with closed eyes, without falling asleep; 8 minutes of fMRI scan without any task (“resting-state”) were acquired for each participant. Additionally, a separate 5-minute long scan was also acquired while a plot-driven story was presented through headphones to participants, who were instructed to listen while keeping their eyes closed. The present analysis will focus on the resting-state data only; the story scan data are not relevant to the work presented here, and will not be discussed further.”

5. Is there any concern that spontaneously ventilating subjects on propofol are hypoventilating and hypercarbic, potentially altering BOLD signals?

We acknowledge the validity of this concern, and we address it in the discussion section in the context of the limitations of our work (see paragraph below).

“Additionally, it is important to consider how the anaesthetic agent can indirectly affect measures of brain activity by altering physiological parameters, such as arterial concentration of carbon dioxide. Since anaesthetized subjects in this study were not intubated but rather spontaneously ventilating, propofol-induced respiratory depression may have resulted in hypoventilation and increased arterial

CO₂ levels. We did not measure end-tidal or arterial CO₂ concentrations in our subjects; however, propofol only has relatively minor effects on brain hemodynamics⁶⁰, and BOLD fluctuation amplitudes, connectivity strength and the spatial extent of connectivity maps have been shown to be unaffected by hypercapnia during the resting state in rats⁶¹, thus mitigating this concern.”

6. While several limitations are addressed in the discussion, more attention and emphasis should be placed on potential limitations of the graph theoretical measures that are utilized here. These limitations are discussed in several recent reviews, such as in Papo et al. Beware of the Small-World Neuroscientist. Frontiers in Human Neuroscience, 2016.

We thank the reviewer for bringing this relevant article to our attention. We have added this point to the Discussion section, citing the paper by Papo et al. regarding limitations of using graph-theoretical measures in brain networks estimated from functional MRI, and explaining how we mitigated this concern in our work (see below).

“It is important to bear in mind that applications of small-worldness and other graph-theoretical measures to brain networks are inherently limited by the noisiness of imaging modalities such as fMRI⁴⁵ – a concern that we sought to address in this work by replicating our results pertaining to small-worldness across multiple thresholds, and with different node and edge definitions for our networks.”

Minor Comments:

1. There are several issues with the references:
 - a. Reference 7 is missing the year, volume, and page numbers.
 - b. Reference 9, 14, 15, and 30 are missing the year
 - c. Reference 19 has an incorrect author list – the first author of this paper is Casali, not Massimini

We thank the reviewer for pointing out these issues: they have been rectified in the revised version of the manuscript.

2. (line 1) "The brain is a remarkably complex system, occupying a near-critical point..." here "occupying" seems to be a poor choice of verbs because it anthropomorphizes and because the brain as a whole is not really near a critical point but maybe its activity patterns are? Not 100% sure if that is a valid criticism or reflection of the criticality notion.

As recommended, we have changed the subject of the sentence, now referring to the brain's activity patterns, and changed the verb. Please see relevant text below.

“The brain is a remarkably complex system, with activity patterns poised at a near-critical point between order and chaos¹, integrating inputs from different modalities into a unified experience of the world.”

3. Line 22 – to avoid inadvertently giving the impression that resting EEG can provide a measurement of the Perturbational Complexity Index, the authors should clarify that these EEG measurements are made after a TMS pulse.

Following the reviewer's remark, this sentence has been re-phrased accordingly, to make it clear that the PCI is based on the EEG evoked response to TMS:

“Studies using a proxy for Φ based on the evoked EEG response to Transcranial Magnetic Stimulation (TMS), known as Perturbational Complexity Index, have been highly successful at discriminating between different states of consciousness, including anaesthesia and disorders of consciousness^{19–22}.”

4. For the sentence that references the Sci Trans Med 2013 paper (ref 19), would recommend citing follow-up papers by Massimini's lab that provide further evidence of the use of the Perturbation Complexity Index as a measure of consciousness: Casarotto et al. Annals of Neurology 2016 and Comolatti et al. Brain Stimulation 2019.

We agree that the articles mentioned are highly relevant: we have therefore incorporated all of them in our manuscript, as doing so will provide the reader with a broader picture of the usefulness and development of PCI.

5. The recent paper by Demertzi et al published in Science Advances is quite relevant to the present paper and should be cited.

The article in question is indeed extremely relevant: it is now cited and discussed in the manuscript, as recommended.

6. (line 83) Baxter A50 (Singapore) should be explained.

We have now clarified in the manuscript that this refers to the infusion pump model.

7. Was a 32-channel coil used for both the rs-fMRI and the T1 MPRAGE sequence? The 32-channel coil is only mentioned in relation to the T1 MPRAGE sequence.

Indeed, the same coil was used for both sequences, and we have now made that clear in the manuscript.

8. The use of anisotropic voxels for the patient T1 sequence (1 x 1 x 1.2 mm) is another minor weakness.

We thank the reviewer for pointing out this issue, which on further inspection turned out to be due to a copy-pasting error from the draft of a different paper: we confirm that the T1 sequences analysed in this study were all 1mm isotropic, for both the propofol and DOC datasets. We have amended the main manuscript to reflect this fact.

9. Table 1 – I assume the authors performed the CRS-R not the CRS – this should be clarified. This is indeed correct: we thank the reviewer for pointing this out, and we have now clarified this in the manuscript, amending the table and its caption.

10. Line 359 – the sentence appears to be missing a verb.

This has now been corrected.

11. There are some redundancies, with repetition of methods in the results section (e.g. lines 464 to 471, and lines 514-526).

The results and methods have been rearranged, to increase readability, following comments from this reviewer 1.

12. There are also interpretation statements in the results that are more appropriate for the discussion (e.g. line 476-477).

The sentence in question has been moved from the Results to the Discussion, as suggested.

13. For the sentence on 510, in addition to references 48-53, consider citing Vanhaudenhuyse et al Brain 2010 and Threlkeld et al Cortex 2018; also consider citing Threlkeld et al on line 675.

These articles are indeed relevant, and including them will provide the reader with a better overview of the literature; thus we have added all of them to the manuscript, as recommended.

14. (lines 591 and 596) Is it correct to say that the global connectivity entropy was "driven" by the changes in the DMN and other networks? Although the mathematical result may have primarily reflected DMN effects, weren't these ultimately just correlated? "Driven" implies a causal relationship such that reduced connectivity entropy in the DMN would cause global connectivity entropy changes, which was not proven in these experiments.

We have changed the phrasing in the manuscript to language that does not imply a causal relationship, since as the reviewer correctly points out, our results did not assess causality.

15. Some of the text in the Supplementary Material is redundant with the main text.

Text has been removed from the Supplementary Information to eliminate redundancy.

16. The layout of supplementary Figures 4-6 is not as intended and does not match their legend.

We apologise for the discrepancy between figures and legends. The seed-based correlation results for DOC and propofol anaesthesia have now been divided into separate figures for simplicity, and the captions have been amended accordingly.

Reviewer #1

Thanks to the authors for addressing all my comments.

Reviewer #2

The authors addressed all my previous points.

Reviewer #3

The revised manuscript was highly responsive to the suggestions and critiques of the original submission, and all of my concerns have been satisfactorily addressed. Readability has also been greatly enhanced by the reorganization.

A minor point: In the new paragraph on "Comparison of Different DOC Aetiologies" the final sentence starts "No significant results were observed ..." but should probably read "No significant differences were observed ..."

Also, the new verbiage on page 12 discussing the Entropic Brain Hypothesis has resulted in too many "Howevers". Perhaps the first one can be removed.

Reviewer #3

The revised manuscript was highly responsive to the suggestions and critiques of the original submission, and all of my concerns have been satisfactorily addressed. Readability has also been greatly enhanced by the reorganization.

A minor point: In the new paragraph on "Comparison of Different DOC Aetiologies" the final sentence starts "No significant results were observed ..." but should probably read "No significant differences were observed ..."

We agree with the reviewer that the suggested formulation is more appropriate; the manuscript has been modified accordingly.

Also, the new verbiage on page 12 discussing the Entropic Brain Hypothesis has resulted in too many "Howevers". Perhaps the first one can be removed.

We have edited the relevant paragraph to remove the first "However", as suggested.